# AuPair: Golden Example Pairs for Code Repair

## Abstract

Scaling up inference-time compute has proven to be a valuable strategy in improving the performance of Large Language Models (LLMs) without fine-tuning. A task that can benefit from such additional inference-time compute is self-repair: given an initial flawed response, the LLM has to correct its own mistake and produce an improved response. We propose leveraging the in-context learning ability of LLMs to perform self-repair. The key contribution of this paper is an approach to synthesise and select a golden set of *pairs*, each of which contains a problem with an initial *guess*, and a consequent *fix*, both generated by the LLM. Each golden example pair, or AuPair[1], is then provided as an in-context example at inference time to generate a candidate repaired solution with 1-shot prompting; in line with best-of-$N$ the highest-scoring response is selected. Given an inference-time compute budget of $N$ LLM calls, our algorithm selects $N$ AuPairs in a manner that maximises complementarity and usefulness. We demonstrate the results of our algorithm on the coding domain for code repair on 4 LLMs across 7 competitive programming datasets. The AuPairs produced by our approach provide a significant boost in performance compared to best-of-$N$, and also exhibit strong generalisation across datasets and models. Moreover, our approach shows strong scaling with the inference-time compute budget.

## 1 Introduction

Recent progress in the field of Large Language Models (LLMs) has resulted in models that keep getting better at generating responses to user queries. When providing these already powerful models with more inference-time compute—increasing number of LLM calls—methods that sample different responses and then select the best among them, such as best-of-$N$ (Stiennon et al., 2020) or self-consistency (Wang et al., 2023b), have shown clear benefits. While these approaches are more breadth-focused, another way to leverage inference time compute is to improve or *repair* the LLM's initial *guesses* by generating better *fixes*. We propose combining the benefits of both these approaches to generate a wide set of repaired solutions for poor initial LLM responses, and then select the best as final answer.

To generate a wide range of repaired solutions for each initial LLM response, we exploit the in-context learning capability exhibited by LLMs. The main contribution of this paper is an algorithm that produces a golden sequence of pairs of *guesses* and *fixes*, which can each be provided as in-context example for generating repaired solutions. Each such AuPair consists of the problem description, the initial guess, and the consequent fix, along with their respective scores. An example AuPair is illustrated in Fig. 2. Given an inference-time compute budget of $N$ LLM calls, our algorithm provides an ordered set of $N$ golden example pairs or AuPairs. These AuPairs are used to generate $N$ fixes at inference time, out of which the highest scoring one is selected as the final output response.

A core ingredient of our proposed algorithm is the selection of these AuPairs. We propose a submodular approach based on the ability of each pair to solve different problems in a held-out validation set. Since the list of AuPairs is constructed by taking the greedy pair at each step, only those pairs that increase the score of the fix on a subset of problems are selected, resulting in *useful* AuPairs.

---

[1]The name AuPair is a coupling of Au, the chemical symbol for gold, and Pair, jointly referring to golden pairs that are produced by our algorithm. The high-level interpretation is that like an "au pair", the approach guides the LLM towards better behaviour.

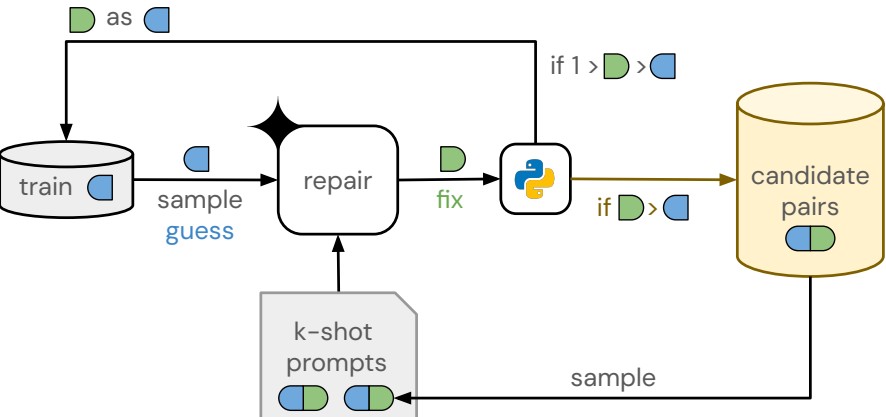

Figure 1: **Pair Generation:** This phase includes collecting a large set $\mathcal{C}$ of guesses and their fixes giving pairs. At each step, a problem with its guess is sampled from the training dataset "train"[left], and used in conjunction with $k$ randomly sampled pairs from the candidate pair buffer to compose a $k$-shot prompt. This prompt is then passed through an LLM to generate a fix. The fix is evaluated on the unit tests by running the Python interpreter and computing its test pass rate. If this fix is better than the guess, this (guess, fix) pair is added to "train". Any improved but imperfect fix is also added as a new guess to the "train" set of guesses. See §2.1 for more details.

Also, as the AuPairs are selected in a submodular manner to solve different sets of problems, by design, we get *complementary* AuPairs. In a nutshell:

*AuPair is a simple and general-purpose selection algorithm, which builds a diverse and useful set of examples that can be provided in context at inference time. It can be used to solve tasks in which the model can repair its own solution to improve performance, provided a grounded source of verification, such as a set of correctness tests.*

In this paper, we focus on the code repair task: given a coding problem, an initial guess which is LLM-generated code, and a set of test cases that are used only to evaluate the correctness of the generated code, can the LLM generate an improved fix for the problem? We show that the fixes generated with AuPairs provided as in-context examples are significantly more *useful* and *diverse* than those generated using best-of-$N$ (§3) for the same inference-time compute budget.

The key contributions of this paper are the following:

- An inference-time **algorithm**, AuPair, which constructs a golden set of code repair examples that boost performance significantly when used as in-context examples (§2).

- **Reliably outperforming** best-of-$N$ across 4 different model sizes: Gemma-9B, Gemma-27B, Gemini-1.5-Flash, Gemini-1.5-Pro, and 7 competitive programming datasets (§3.1).

- Strong **scaling** performance with inference time compute, with far less diminishing returns than best-of-$N$ (§3.3).

- Robust out-of-distribution **generalisation**, w.r.t. both model size and dataset (§3.4).

- Demonstrably higher **diversity** of solutions, without performance trade-off (§3.6).

## 2 APPROACH

The goal of our algorithm is to improve code repair performance on unit tests at inference time, by building a list of pairs that can be provided as in context examples. The code repair prompt includes an optional set of examples, followed by a text description of the problem to solve and the initial *guess* generated by the LLM. The LLM generates a revision, or a *fix* that improves performance on the unit tests for that problem, see Fig. 2. In the prompt, we also include the scores achieved by the

Figure 2: **An example AuPair** ◖: guess/fix from CodeForces and their respective test pass rates [above], and the problem description [below]. The guess checks only the first digit for every single number leading up to the input. The fix corrects the logic by iterating over the *divisors* of the input, and checking for an intersection over *all* digits with the input.

guess and fix on the unit tests, but no additional execution feedback.[2] Our approach consists of two main phases: 1) Pair Generation §2.1, and 2) AuPair Extraction §2.2.

In order to disentangle repair performance from the quality of initial guesses, we first curate composite datasets consisting of initial guesses for all the coding problems. Given a dataset consisting of problems and their corresponding tests, we first generate an initial *guess* for each problem and compute its score on the unit tests. If the guess passes all the unit tests for that problem correctly, no further improvement is required and we discard that problem. If not, we add this guess along with its corresponding score and problem as a datapoint to our curated dataset. This dataset is then divided into training, validation, and test datasets. We use the training dataset $\mathcal{D}_{train} \equiv \mathcal{D}$ for pair generation (Fig. 1), and the validation dataset $\mathcal{D}_{val}$ for AuPair extraction. The test dataset is used in the final testing phase only $\mathcal{D}_{test}$.

## 2.1 PHASE 1: PAIR GENERATION

In this phase, we generate a set $\mathcal{C}$ of candidate (guess, fix) pairs using the approach illustrated in Fig. 1. These pairs will then be used to select the AuPairs in the next phase. For each problem sampled from the training dataset $\mathcal{D}$, we have an initial guess. Next, the LLM has to generate a fix for this guess. To collect a wide variety of fixes, we randomly sample $k$ pairs from the existing set of candidate pairs $\mathcal{C}$ and provide them as in-context examples of code repair. Note that initially the candidate pair buffer is empty so there will be no in-context examples. However, this candidate pair set $\mathcal{C}$ gradually gets populated as more fixes are generated by the LLM. These $k$ example pairs, along with the problem and its initial guess, are used to compose a $k$-shot repair prompt. This repair prompt is then provided as input to the LLM, which generates a fix that is scored on the unit tests. If this score is an improvement over the guess score, this (guess, fix) pair is added to the set of candidate pairs. Furthermore, if this fix is imperfect, i.e., it does not pass all the test cases, it

---

[2]The repair prompt is composed using the prompting strategy shown in Fig. A.3.

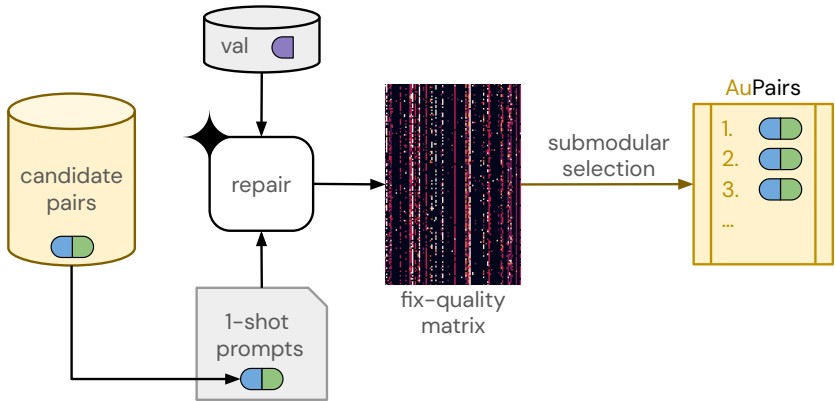

Figure 3: **Au**Pair Extraction: given a large set $\mathcal{C}$ of "candidate pairs" ⬤, each pair is provided as a 1-shot in-context example in the prompt for each problem and its guess ⬤ from the validation set "val". These prompts are passed to the LLM which generates one fix at a time to all the guesses in the validation set "val". These fixes are evaluated on the corresponding unit tests to populate a "fix-quality matrix" $M \in \mathbb{R}^{|\mathcal{C}| \times |\mathcal{D}_{\text{val}}|}$, as described in **Algorithm 1**. Then, a submodular selection mechanism is applied to obtain the list of AuPairs, in **Algorithm 2**, see §2.2 for details.

becomes a potential guess with further scope for improvement, so we add it as a guess to our training dataset $\mathcal{D}$. This process is repeated several times to collect a large set of such candidate pairs. [3]

## 2.2 PHASE 2: AuPAIR EXTRACTION

Now that we have a large set $\mathcal{C}$ of candidate pairs, the next step is to determine which of these will actually help boost performance, i.e., which of these are AuPairs. We do this in a *submodular* fashion by making use of the validation dataset $\mathcal{D}_{\text{val}}$. For every single pair-problem combination $(\boldsymbol{c}_i, \boldsymbol{x}_j) \in \mathcal{C} \times \mathcal{D}_{\text{val}}$, we build a 1-shot prompt $\boldsymbol{p}$ using the prompting strategy described in A.3 to query the LLM to generate a fix for the given problem $\boldsymbol{x}_j \in \mathcal{D}_{\text{val}}$. The fix generated by the LLM is then evaluated on the unit tests and stored in the fix quality matrix $M \in \mathbb{R}^{|\mathcal{C}| \times |\mathcal{D}_{\text{val}}|}$ at index $(i, j)$. This first step of AuPair extraction is outlined in **Algorithm 1**.

---

**Algorithm 1** Fix quality matrix computation

**Require:**
$\begin{cases} \text{LLM} & \text{large language model} \\ \mathcal{C} & \text{candidate pairs} \\ \mathcal{D}_{\text{val}} & \text{validation dataset} \\ \text{score} & \text{code eval function} \end{cases}$

1: init fix quality matrix $M \leftarrow \mathbf{0}^{|\mathcal{C}| \times |\mathcal{D}_{\text{val}}|}$
2: **for** pair $\boldsymbol{c}_i$, problem $\boldsymbol{x}_j \in \mathcal{C} \times \mathcal{D}_{\text{val}}$ **do**
3:     build 1-shot prompt: $\boldsymbol{p} \leftarrow \boldsymbol{c}_i \,\|\, \boldsymbol{x}_j$
4:     generate fix: $\hat{\boldsymbol{y}} \leftarrow \text{LLM}(\boldsymbol{p})$
5:     evaluate fix: $M_{i,j} \leftarrow \text{score}(\hat{\boldsymbol{y}})$
6: **end for**
   return $M$

---

**Algorithm 2** Submodular AuPair extraction

**Require:**
$\begin{cases} M & \text{fix quality matrix} \\ \mathcal{C} & \text{candidate pairs} \\ \epsilon & \text{tolerance} \end{cases}$

1: initialise AuPairs $\mathcal{A} \leftarrow ()$
2: **repeat**
3:     per-pair scores: $\bar{\boldsymbol{m}} \leftarrow \text{row-mean}(M)$
4:     get best pair: $\boldsymbol{c}_k \leftarrow \text{argmax}_{\mathcal{C}} \, \bar{\boldsymbol{m}}$
5:     append to AuPairs: $\mathcal{A} \leftarrow \mathcal{A} \cup \boldsymbol{c}_k$
6:     update $M \leftarrow \text{clip}(M - M_k, 0, 1)$
7: **until** $\max(\bar{\boldsymbol{m}}) < \epsilon$
   return $\mathcal{A}$

---

Next, we use this fix quality matrix $M$ to extract the AuPairs by taking the following steps: 1) Select the pair that gets the highest mean score across all problems in $\mathcal{D}_{\text{val}}$, say $\boldsymbol{c}_k$, and add it to the list of AuPairs $\mathcal{A}: \mathcal{A} \leftarrow \mathcal{A} \cup \boldsymbol{c}_k$. This is a greedy way of selecting the best pair given all previous AuPairs and produces an ordered set of AuPairs. 2) Subtract the row score $M_k$ (i.e. score on all the problems in $\mathcal{D}_{\text{val}}$) of this newly added pair from all the rows in the fix quality matrix with an update: $M - M_k$. This ensures that redundant AuPairs are not produced by the approach. The updated

---

[3]Please refer to Table 2 for initial candidate pair buffer details.

fix quality matrix is clipped to $(0, 1)$ since any negative value in the matrix $M$, say $M_{i,j}$, implies that the problem $x_j$ cannot be improved further by pair $c_i$. Without clipping, we would not get an accurate estimate of the improvement in the next step of submodular extraction. 3) this process is repeated till the improvement falls beyond a tolerance $\epsilon$. This submodular extraction of AuPairs is shown in **Algorithm 2**. Fig. 3 has a joint diagram depicting fix quality matrix computation and submodular AuPair extraction.

This process of iteratively constructing the set of AuPairs ensures that they improve performance on disjoint parts of the problem space. The AuPairs that we obtain from this phase are then used in the same manner at inference time, as 1-shot examples, to improve code repair performance. The compute budget $N$ at inference time determines the number of AuPairs that we can use at inference time. Since the AuPairs form an ordered set, the first $N$ AuPairs are used at inference time for budget $N$. The final solution for each problem is the best among all generated solutions, i.e., the one that passes the most test cases.

## 3   Experiments

**Datasets:** We use 7 datasets that contain problems and test cases from competitive programming contests: 1) CodeForces (8.8k problems), 2) AtCoder (1.3k problems), 3) HackerEarth (1.2k problems), 4) CodeChef (768 problems), 5) LiveCodeBench (400 problems), 6) CodeJam (180 problems), and 7) Aizu (2.2k problems) (Li et al., 2022a; Jain et al., 2024). We choose Code-Forces and AtCoder, separately, for in-distribution testing, and use the rest exclusively for out-of-distribution testing. Our train / val / test split proportions for the CodeForces and AtCoder datasets are $37.5/12.5/50\%$. Some datasets have difficulty levels as part of the problem; for those datasets we maintain the same stratified distribution of questions in the training, validation, and test datasets.

**Models:** We use 4 models of different sizes: Gemma-9B, Gemma-27B, Gemini-1.5-Flash and Gemini-1.5-Pro. In addition to using these models for dataset curation and pair generation, we look at the transfer capabilities of our method with respect to different models in Section 3.5.

**Evaluation:** We use each AuPair as 1-shot example, in context with the problem text and initial guess in the repair prompt. The structure of the prompt is the same as the one used earlier (A.3). We perform two types of evaluation: in-distribution and out-of-distribution. For in-distribution evaluation, we use the test split from the same dataset as the one used for pair generation and AuPair extraction. This ensures that the format of questions and test cases in the test questions matches that of the AuPairs. Out-of-distribution evaluation uses a different coding dataset; this means that the test samples have different format of questions, difficulty, types of problems and test cases than the AuPairs. Another axis of out-of-distribution evaluation that we look at is the model axis: we report the performance obtained using AuPairs produced by a different model than the one used at inference time.

**Metrics:** Our primary metric is the best-of-$N$ accuracy, which we calculate as the average of the best response across $N$ LLM calls for all points in the test dataset. In our case, we have grounded feedback available in the form of the number of unit tests passed by each LLM response, when executed using the Python interpreter. We use this feedback to compute the maximum score out of the $N$ generated outputs for our approach and baselines, all of which involve $N$ LLM calls at inference time for fair comparison. For our approach we pick the first $N$ AuPairs from $\mathcal{A}$.

**Baselines:** We compare the effectiveness of our proposed approach with best-of-$N$ (Stiennon et al., 2020) and self-repair (Olausson et al., 2024). Best-of-$N$ is currently the strongest baseline to improve model performance by allowing multiple ($N$) LLM calls at inference time. To have a strong best-of-$N$ baseline, we set the temperature to 1.0, to ensure sampling of diverse responses while preserving good quality responses (Renze & Guven, 2024) Self-repair uses the $N$ LLM calls to either generate verbal feedback or repaired code. In our experiments, for a budget of $N = 32$ LLM calls, we use 4 LLM calls to generate verbal feedback and 7 LLM calls to generate repaired code for each verbal feedback.

The remainder of this section will discuss a plethora of empirical results, on overall and ablated performance (§3.1 and 3.2), scalability and generalisation (§3.3 to 3.5), and diversity (§3.6 to 3.8).

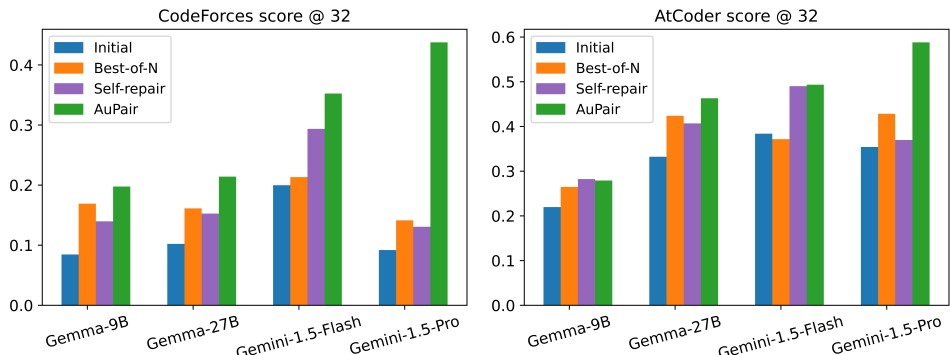

Figure 4: **In-distribution code repair performance:** with $N = 32$ LLM calls at inference time and the same train / val / test data distribution, we compute the average pass rate on test cases. The same model is used for generating the initial guesses and fixes and the AuPair extraction. CodeForces (left, 8.8k problems) and AtCoder (right, 1.3k problems), see §3.1 for more details.

## 3.1 SIGNIFICANTLY BOOSTED CODE REPAIR PERFORMANCE

The first step to assess code repair performance is to measure *in-distribution* performance; namely generate and selecting AuPairs on the training and validation sets that match the test dataset, and using the same model at evaluation as for construction. We do this for 2 datasets (CodeForces and AtCoder) and all 4 models. Fig. 4 shows the resulting comparison between the best-of-$N$ baseline and our AuPair approach, for a budget of $N = 32$ LLM calls at inference time.[4] AuPair is clearly superior to best-of-$N$ on all models and datasets, sometimes by wide margins. This clearly establishes that our proposal of providing a different in-context example of code repair in each LLM call can significantly boost performance.

An interesting side-result is visible in initial performance, i.e., the performance of the initial responses of the LLMs to the problems, which have to then be repaired. Gemini-1.5-Pro, despite being a superior model to Gemini-1.5-Flash, shows worse initial performance. Since the code generated has certain conditions that allow successful execution, we observe that many initial guesses of generated code fail because they do not obey these conditions (see Appendix §A.4). In such cases, code repair with best-of-$N$ is unlikely to give us high boost in performance since the initial solution is badly formatted. This is one clear case where having an AuPair in context significantly improves performance. As a result, using AuPairs in conjunction with high performing models leads to large performance improvements despite poor initial performance, as we can see for both CodeForces and AtCoder with the Gemini-1.5-Pro model in Fig. 4.

## 3.2 SELECTION MATTERS: AUPAIRS ARE MORE EFFECTIVE THAN RANDOM PAIRS

We design an ablation to disentangle the two possible sources of improvement that our approach demonstrates, namely 1) in-context learning and 2) the choice of AuPairs. It is not implausible for the boost in performance to result from the LLMs' in-context learning ability, and that the same result could be achieved by including *any* set of pairs. On the other hand, our approach specifically targets complementarity during construction of AuPairs in that subsequent AuPairs are selected based on their ability to solve problems that previous AuPairs were unable to solve. To resolve this, we compare the full method to a random-pair baseline that randomly selects pairs from the full candidate set (the result of Phase 1), deduplicating the problems that the random pairs solve (which makes it a stronger baseline). Fig. 5 shows that AuPair significantly outperforms the random-pair baseline for $N = 1, ..., 32$. Note that for any fixed candidate set, as $N$ grows toward the size of the full set of pairs, the performance of the random-pair baseline will equal that of AuPair.

---

[4]Since our algorithm yields a variable number of AuPairs, for smaller datasets with fewer generated pairs, the total number of AuPairs can be less than 32. To have a fair comparison in that case, we set the same compute budget $N$ for the best-of-$N$ baseline. This is the case for AtCoder (Fig. 4, right), where our algorithm yields 14 and 27 AuPairs for Gemma-9B and Gemma-27B respectively. So the corresponding best-of-$N$ baseline results also use a matching compute budget of 14 and 27 LLM calls respectively.

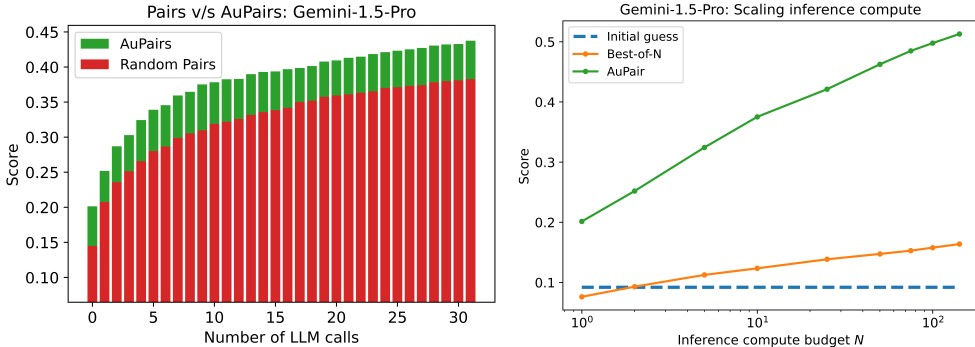

Figure 5: **(a) AuPairs vs. random pairs**: AuPairs (green) are significantly (about $3\times$) more compute efficient than random pairs (red); it takes only 11 AuPairs to reach the same performance as 32 random pairs (CodeForces dataset, Gemini-1.5-Pro); **(b) Scaling inference-time compute:** using AuPairs the score increases with compute budget at a much steeper rate compared to best-of-$N$.

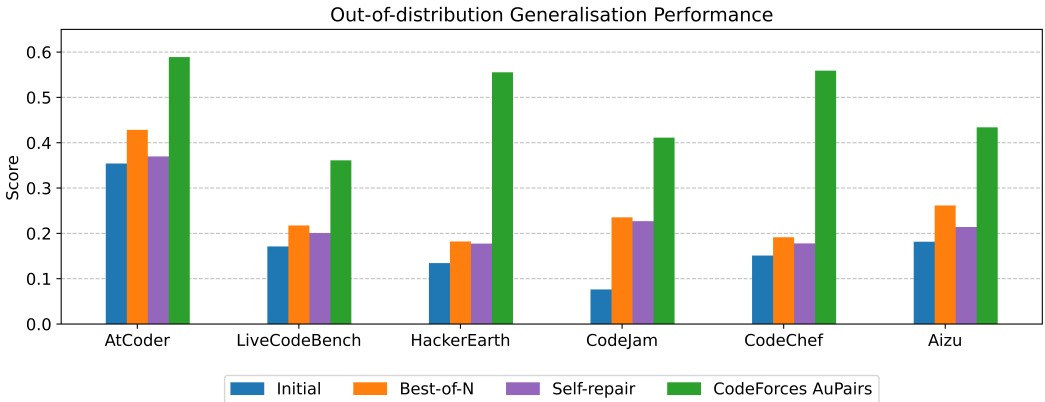

Figure 6: **Out-of-distribution code repair performance:** AuPairs extracted on the CodeForces dataset show strong generalisation performance across the other six datasets with Gemini-1.5-Pro.

### 3.3 BETTER SCALING WITH INFERENCE-TIME COMPUTE

At a fixed budget of $N = 32$ LLM calls, our results look promising. In this section, we investigate whether and how performance scales with $N$. Fig. 5(b) plots the score as a function of the inference compute budget $N$ using Gemini-1.5-Pro (additional scaling curves in the Appendix, see Fig. 9). For each additional LLM call, we use the next best AuPair produced by the algorithm and provide it in context to generate the LLM response. The results shows a clear scaling trend with a consistent log-linear performance increase as a function of compute, without any sign of a plateau. More importantly, the increase is substantially *steeper* than for the best-of-$N$ baseline; in other words, our prompting with complementary AuPairs makes more efficient use of compute than repeated sampling given a fixed prompt.

### 3.4 STRONG GENERALISATION TO OUT-OF-DISTRIBUTION DATASETS

The aim of this set of experiments is to determine whether our approach exhibits out-of-distribution generalisation, i.e., given AuPairs collected on a different dataset, see if we can retain the performance improvements that we obtain in-distribution. We evaluate the AuPairs collected using the Gemini-1.5-Pro model on the CodeForces dataset on the other 6 datasets and compare them with the corresponding best-of-$N$ baselines. Fig. 6 shows that for all 6 datasets, our approach outperforms best-of-$N$ by a large margin, in spite of having out-of-distribution AuPairs. This in turn implies that the process of collecting AuPairs may only be needed on one dataset, and its benefits can be reaped across a wide range of problems (from other datasets, or users) at inference time.

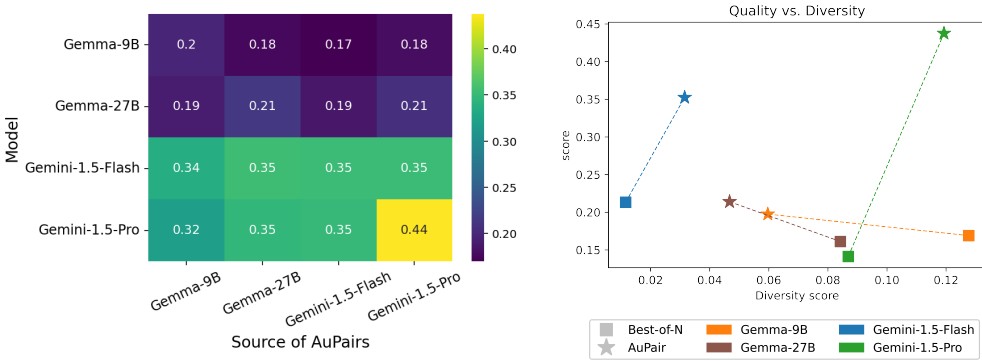

Figure 7: **(a) Cross-model transfer:** AuPair shows good cross-model transfer capabilities for all four models on CodeForces; **(b) Diversity-Score plot:** we calculate diversity as the percentage of unique subtrees present in Abstract Syntactic Trees of the $N$ generated fixes (higher is better). AuPair ($\star$) with Gemini-1.5-Flash and Gemini-1.5-Pro generates more diverse responses than best-of-$N$ ($\square$) while this diversity trend is reversed for the Gemma models. In terms of score, AuPair always generates higher-scoring fixes than best-of-$N$.

### 3.5 DECENT CROSS-MODEL TRANSFER

Now that we have seen that our approach can exhibit very good out-of-distribution generalisation along the data axis, we evaluate it on its ability to generalise on the model axis, i.e., we look at the performance of AuPairs collected using a different model. We evaluate this cross-model transfer capability for all model combinations on CodeForces. The resulting 16 ablations are shown in Fig. 7(a), and help disentangle the impact of the AuPairs versus the code repair capabilities of the inference model. A key takeaway is that the Gemma models exhibit worse performance, regardless of the quality of AuPairs used at inference time, indicating that they are inferior at the capability of code repair. Gemini-1.5-Flash performs much better at code repair, and its sensitivity to the source of AuPairs is negligible: it is equally performant for each source. Gemini-1.5-Pro, on the other hand, *is* sensitive to the source of AuPairs; in particular, when Gemini-1.5-Pro uses AuPairs collected by the same model, it achieves the best performance by a large margin. With AuPairs selected using other models, Gemini-1.5-Pro achieves comparable performance to Gemini-1.5-Flash. One reason for the standout performance when using Gemini-1.5-Pro AuPairs seems that those examples result in substantially more diverse generations, as shown in Section 3.6. However, Fig. 7(a) as a whole suggests that there is an ordering in terms of performance: 1) the model used at inference time has to have good code repair capabilities, and 2) the stronger the model is at code repair, the more improvement we can expect from it with a higher *quality* of AuPairs.

### 3.6 HIGH CODE-SPECIFIC DIVERSITY

We dive a bit deeper into the nature of fixes generated using different AuPairs. There are several ways to analyse code; we choose Abstract Syntax Trees (ASTs) since they mostly capture the structure of changes. More concretely, since we have $N$ fixes for each problem (here $N = 32$), we measure the diversity per problem as the number of unique changes made to the guess over all $N$ fixes for that problem. The diversity score is calculated as the average number of unique abstract syntactic subtrees generated per problem. More concretely, we perform the set difference of all subtrees in the fix AST that are not in the guess AST and normalize with the maximum number of subtrees. We plot this diversity metric against the score in Fig. 7(b) to get a sense of how diverse and useful the AuPairs are. We also include diversity results of the best-of-$N$ baseline, see A.2 for further details on the diversity score computation. The results show that while AuPairs always increase performance, they result in higher diversity of fixes when given to the more competent models (Gemini-1.5-Pro and -Flash), and lower diversity for Gemma models. It is worth highlighting that the exceptional performance of AuPairs produced and used by Gemini-Pro (Fig. 7(a), bottom right) correspond to highly diverse fixes (Fig. 7(a), top right).

| Difficulty level → | A (671) | B (675) | C (671) | D (666) | E (649) | F+ (537) |
|---|---|---|---|---|---|---|
| Gemma-9B | 0.34 (+0.16) | 0.23 (+0.13) | 0.19 (+0.12) | 0.15 (+0.09) | 0.14 (+0.08) | 0.12 (+0.07) |
| Gemma-27B | 0.28 (+0.1) | 0.25 (+0.12) | 0.20 (+0.12) | 0.19 (+0.1) | 0.17 (+0.1) | 0.20 (+0.11) |
| Gemini-1.5-Flash | 0.54 (+0.2) | 0.39 (+0.18) | 0.34 (+0.15) | 0.18 (+0.11) | 0.26 (+0.12) | 0.28 (+0.11) |
| Gemini-1.5-Pro | 0.62 (+0.42) | 0.52 (+0.4) | 0.43 (+0.35) | 0.38 (+0.32) | 0.32 (+0.28) | 0.35 (+0.29) |

Table 1: **Difficulty-wise analysis:** score (§3) using AuPairs, categorised by difficulty level from easy (A) to hard (F+), accompanied by number of problems. Absolute improvement in parentheses. We see an expected trend here: the strongest performance is observed using the best models on the easiest problems, and as difficulty increases, performance decreases across models. However, our results with Gemini-1.5-Pro indicate improved performance with higher difficulty

### 3.7    Improvement on All Difficulty Levels

Coding datasets have heterogeneous difficulty. As a sanity check, we conduct additional analysis to determine *which problem levels* are most helped by AuPair, compared to the quality of initial guesses. Table 1 shows the absolute improvement in score, i.e., the increase in score achieved by AuPair for all 4 models on CodeForces. The two key observations are (a) AuPair helps significantly at all difficulty levels for all models, and (b) there are larger improvements on easier levels, and this trend is consistent across models. Note that the initial performance of Gemini-1.5-Pro is low because the initial guesses generated do not adhere to the instruction (elaborated in Appendix §A.4); however since this is the strongest model and shows the best overall performance across difficulty levels, the increases in score that we see are significantly higher than the other models.

### 3.8    Coverage of Problem Categories is Preserved

The CodeForces dataset is richly annotated with category labels for each problem. A problem may have multiple tags, for instance, `strings` and `two pointers`. We use these fine-grained tags to study how the problem distribution is affected by Phase 1 and Phase 2 of our method, separately. Fig. 8 shows the proportions of these categories observed in the initial dataset, the full set of pairs generated during Phase 1, and the final AuPairs. The high-level result is encouraging, namely that the starting diversity is approximately preserved. Phase 1 yields pairs for every single category, even those that lie at the tail. Furthermore, the (sparser) distribution over categories for the AuPairs after Phase 2 still shows several problems from rare categories. This additional result consolidates our insight that AuPairs are highly diverse, also in the types of problems they contain.

## 4    Related Work

Automatic Program Repair (APR) has been a longstanding research area in the field of machine learning (Devlin et al., 2017; Bhatia & Singh, 2016; Chen et al., 2019; Feng et al., 2020; Berabi et al., 2021; Chakraborty et al., 2022; Yuan et al., 2022). Most methodologies rely on supervised finetuning to adapt LLMs to the task of code generation using labeled pairs of broken / fixed code pairs, which is costly to obtain and often task- and problem-specific (Hu et al., 2022; Jiang et al., 2021; Xia & Zhang, 2022; Dinella et al., 2020). On the other hand, unsupervised APR is challenging since it requires syntactic and semantic understanding of code, and most automatic code breaking approaches tend to be out-of distribution with real samples. Yasunaga & Liang (2021) train both a breaker and a fixer in order to learn to propose new code fixes that are realistic, and uses a compiler to verify its correctness. Close to our approach we use partial fixes generated by the model as the initial broken code to be fixed iteratively.

More recently, a few unsupervised approaches have been proposed based on the capability of LLMs to generate code (Chen et al., 2021; Nijkamp et al., 2023; Chowdhery et al., 2024; Li et al., 2022b; Fried et al., 2023; Li et al., 2023). The APR task still remains challenging, even though models are better at generating code (Olausson et al., 2024; Chen et al., 2023). Zhao et al. (2024) use a step-by-step method to repair code using a reward model as a critic, providing feedback to finetune an LLM. Shypula et al. (2024) propose a retrieval based few-shot prompting approach with Chain-of-Thought (CoT) reasoning traces, and use supervised fine-tuning (SFT) to finetune a model using self-play.

The main disadvantage of using SFT approaches comes from the need to finetune the model to the task, which becomes much more costly with ever-growing model sizes. In recent years the in-

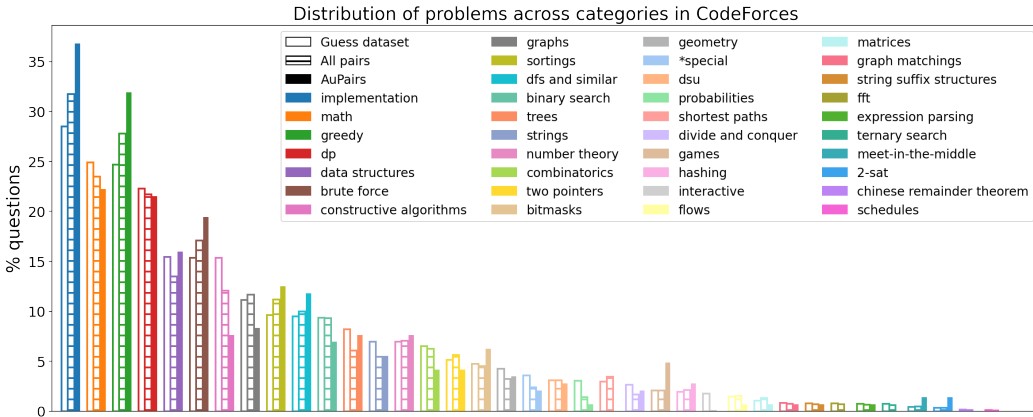

Figure 8: **Category-wise analysis:** analysing the distribution of AuPairs across different categories and comparing it with the distribution of problems in the dataset.

context learning (ICL) paradigm (Brown et al., 2020) has been shown to be a flexible and compute efficient adaptation approach to new tasks (Von Oswald et al., 2023; Akyürek et al., 2023). Le et al. (2022) use an LLM to generate code and a critic network to predict functional correctness of the the generated program, with zero-shot transfer to new tasks. Our work focuses on tasks which the correctness is specified by the number of test cases the generated code passes. Gou et al. (2024) combine the use of LLMs with tools to provide feedback for the LLM to self-correct via additional calls to evaluate its own output in a validation setting. Wang et al. (2023a) also make use of external tools and use an LLM in a learner / teacher role to provide a chain of repairs to fix the code.

Yin et al. (2024) propose an automated self-repair approach with few-shot prompting but using CoT and execution feedback information. Agarwal et al. (2024) also use CoT rationales but remove them from context when few-shot-prompting the model. Olausson et al. (2024) show that using an LLM as a feedback source for self repair has its limitations when compared with the same number of independent model calls for the same problem since the ability to generated better code may be interconnected with the ability to identify its faulty behaviour. Welleck et al. (2023) decouple the generation and the correction phase, by independently training a corrector with scalar and natural language feedback to correct intermediate imperfect generations. We use self-corrections, since we use the same model for generating the fixes and the broken code pairs, but the improvement is grounded on the number of passing tests, avoiding degenerate behaviours.

Yuan & Banzhaf (2017) propose a multi-objective evolutionary algorithm to search over possible correct code patches; Romera-Paredes et al. (2023) use an island-based evolutionary method to encourage exploration of diverse programs, and perform iterative best-shot-prompting to improve the quality of the generated code. In this paper, we use a generative approach; closer to the work of Shirafuji et al. (2023), we make use of ICL abilities of LLMs to generate improved code repairs, but we provide an extra submodular process to select the samples, that encourages diversity.

## 5 CONCLUSIONS AND FUTURE WORK

We propose an algorithm, AuPair, which produces a set of golden example pairs that can be provided as in-context examples using 1-shot prompting to improve code repair performance at inference time. Our approach is highly scalable, showing significantly better outcomes than best-of-$N$, which is the current state-of-the-art method that improves performance as inference compute is scaled up. In addition to this, the AuPairs generated using our algorithm show strong out-of-distribution generalisation and thus can be reused at inference time to solve a wide range of problems. While in this paper we have explored repair in the coding domain, our algorithm is general and can be used in any setting in which an initial solution generated by an LLM can be improved via repair. Additionally, the choice of coding implies that all our feedback is grounded, but our algorithm is general enough to accommodate ungrounded feedback from reward models, so future work merging this line of research into the full LLM pipeline might be worth investigating.

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

# A   APPENDIX

## A.1   PAIR GENERATION

In this section, we discuss the specifics of the pair generation phase and provide results pertaining to this phase. The approach that we use for pair generation is provided in Algorithm 3. Note that this is one way to generate pairs; they can be generated in other ways, or be available beforehand.

---
**Algorithm 3** Pair Generation

---
**Require:** $\begin{cases} \text{LLM} & \text{large language model} \\ \mathcal{D}_{\text{train}} & \text{training dataset} \\ k & \text{number of few-shot examples} \\ N & \text{total number of LLM calls} \\ \text{score} & \text{code eval function} \end{cases}$

1: init candidate pairs $\mathcal{C} \leftarrow \{\}$
2: **for** $i = 1, \ldots, N$ **do**
3:     sample problem from dataset: $\boldsymbol{x} \sim \mathcal{D}_{\text{train}}$
4:     sample $k$ pairs to use in-context: $\boldsymbol{c}_1, \ldots, \boldsymbol{c}_k \sim \mathcal{C}$
5:     build $k$-shot prompt: $\boldsymbol{p} \leftarrow \boldsymbol{c}_1 \parallel \ldots \parallel \boldsymbol{c}_k \parallel \boldsymbol{x}$
6:     generate fix: $\hat{\boldsymbol{y}} \leftarrow \text{LLM}(\boldsymbol{p})$
7:     evaluate fix: $s_{\hat{\boldsymbol{y}}} \leftarrow \text{score}(\hat{\boldsymbol{y}})$
8:     **if** $s_{\hat{\boldsymbol{y}}} > s_{\boldsymbol{x}}$ **then**
9:         create new pair: $\boldsymbol{c} \leftarrow \langle \boldsymbol{x}, \hat{\boldsymbol{y}} \rangle$
10:         add to candidate pairs: $\mathcal{C} \leftarrow \mathcal{C} \cup \boldsymbol{c}$
11:         **if** $s_{\hat{\boldsymbol{y}}} < 1$ **then**
12:             create new problem $\hat{\boldsymbol{x}}$ with guess $\hat{\boldsymbol{y}}$
13:             add new problem to dataset: $\mathcal{D}_{\text{train}} \leftarrow \mathcal{D}_{\text{train}} \cup \hat{\boldsymbol{x}}$
14:         **else**
15:             remove problem from dataset: $\mathcal{D}_{\text{train}} \leftarrow \mathcal{D}_{\text{train}} - \{\boldsymbol{x}\}$
16:         **end if**
17:     **end if**
18: **end for**
    **return** $\mathcal{C}$

---

### A.1.1   RESULTS

We report the results obtained after phase 1 of our algorithm, pair generation. For the AtCoder dataset, we set a budget of 10,000 LLM calls for pair generation. Since the CodeForces dataset is larger, we set a budget of 35,000 LLM calls to maintain a good balance between having enough LLM calls per problem and maintaining the affordability of the overall approach in terms of computational resources. We report the number of pairs generated on both of these datasets across all 4 models: Gemini-1.5-Pro, Gemini-1.5-Flash, Gemma-27B, and Gemma-9B in Table 2. Here we provide some additional results that we were unable to include in the main text.

| CodeForces | # of pairs | # AuPairs |
|---|---|---|
| Gemini-1.5-Pro | 1560 | 144 |
| Gemini-1.5-Flash | 1327 | 110 |
| Gemma-27B | 509 | 77 |
| Gemma-9B | 556 | 122 |

| AtCoder | # of pairs | # AuPairs |
|---|---|---|
| Gemini-1.5-Pro | 927 | 64 |
| Gemini-1.5-Flash | 397 | 64 |
| Gemma-27B | 295 | 27 |
| Gemma-9B | 147 | 14 |

Table 2: Number of pairs collected during phase 1 of the algorithm (# of pairs) and number of AuPairs extracted in phase 2 (# AuPairs): CodeForces (top) and AtCoder (bottom) for all 4 models.

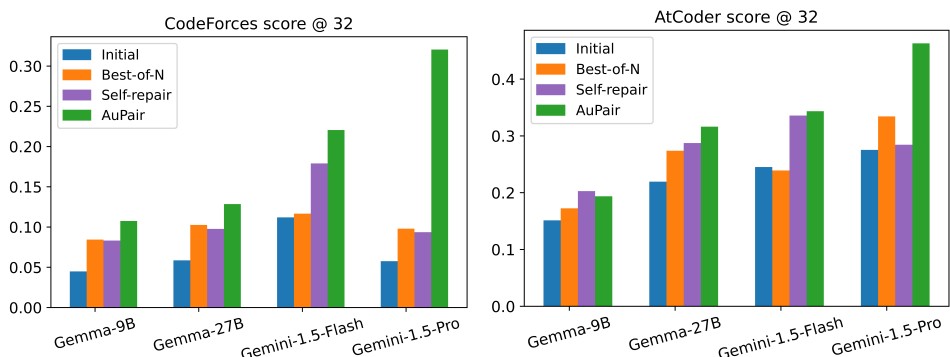

Figure 10: **Percentage of fully solved problems** with $N = 32$ LLM calls at inference time. Code-Forces (left) and AtCoder (right).

### A.1.2 SCALING INFERENCE COMPUTE

In addition to the scaling experiment we performed using Gemini-1.5-Pro, results in Fig. 5(b), we also perform the same scaling experiment using Gemini-1.5-Flash and show the results in Fig. 9. The trend is similar to what we observed before: best-of-$N$ plateaus after a certain number of LLM calls, while our approach scales as the compute budget increases, delivering an improvement in performance for each newly included AuPair. Since our AuPairs are selected submodularly, the initial pairs yield high returns in performance and these returns start diminishing slowly, but notably, performance does not plateau yet. Thus, it is abundantly clear that using AuPairs has a distinct advantage over current state-of-the-art approaches like best-of-$N$ in improving performance at inference time as compute budget increases.

### A.1.3 MEASURING CORRECTNESS IN TERMS OF SOLVED PROBLEMS

In addition to pass rate of unit tests, we also report the percentage of fully solved problems, for which the generated code passes all test cases. We see that AuPair outperforms all other baselines on all models across the board, with results for CodeForces and AtCoder shown in Fig. 10.

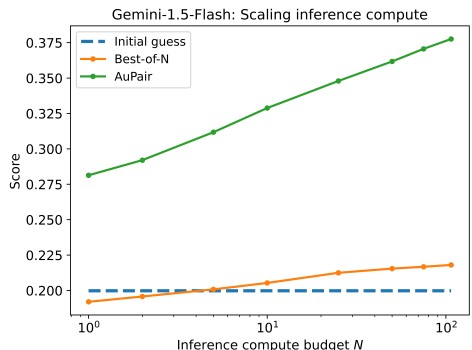

Figure 9: Scaling up inference compute on the CodeForces dataset with Gemini-1.5-Flash. Scores correspond to average pass test rate on all the test problems. *blue dashed line* represents the average score of the initial guesses. *orange* show the best of $N$ x 1-shot prompt, *green* show the best of $N$ AuPair 1-shot prompt. With increasing compute $N$ we can see a clear improvement with Aupair prompting on a much larger steeper slope than best-of-N.

### A.1.4 CODE REPAIR WITH LIVECODEBENCH

Generalisation of AuPair prompting is important to improve code repair of smaller datasets. We posit that the AuPairs contain diverse code changes that transfer meaningfully across datasets, which may be important to those with scarce data.

We now show some examples of AuPairs obtained for a smaller dataset (400 problems) Live-CodeBench (LCB) (Jain et al., 2024). We generated the same train/val/test split (37.5/12.5/50%) over 400 problems and applied our AuPair approach to obtain in distribution AuPairs for LCB.

Fig. 11 shows that even with smaller number of selected AuPairs we still obtain a gain over best-of-$N$ prompting. We obtained 5 AuPairs with the submodular extraction in Algorithm 2 for all the

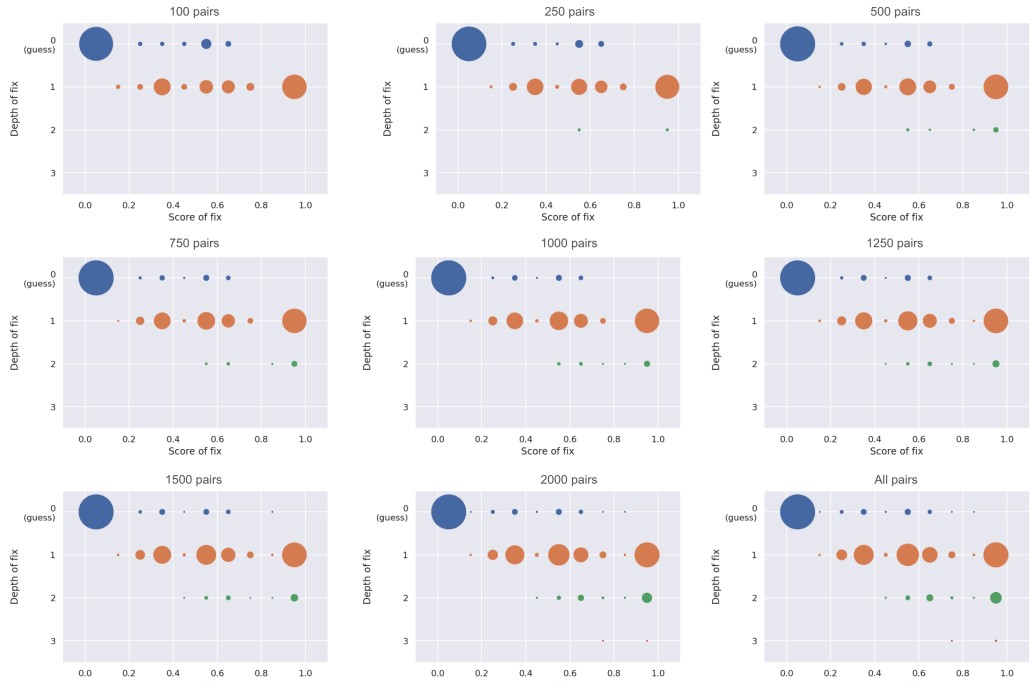

Figure 12: Visualising the lineage of the set of all pairs as the first phase of the algorithm, pair generation, progresses.

models except Gemma-9B which obtained only 3 AuPairs. Given the difference in dataset size these values are larger in proportion to the ones obtained from a larger dataset CodeForces (8.8k problems, 144 extracted AuPairs).

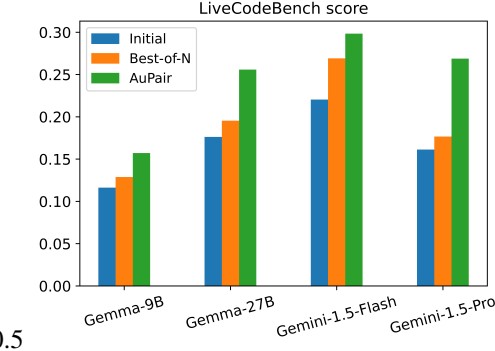

Figure 11: **LiveCodeBench in distribution results** show AuPair prompting is outperforming best-of-N even in the small data regime.

### A.1.5 LINEAGE

Here we look at the lineage of each pair generated during phase 1 of our algorithm, pair generation. The key idea here is to see if the set of all pairs collected during the pair generation phase are *deeper* i.e., they generate iteratively better solutions for a smaller set of problems, or *broader* i.e., they generate solutions for a larger set of problems but those solutions may not necessarily be perfect. The last plot in Fig. 12 (pairs generated on the CodeForces dataset using Gemini-Pro-1.5) indicates that the pairs collected have shallow lineage: a large proportion of guesses that had a score of 0 had corresponding fixes with perfect score at depth 1. We also see that the number of fixes decreases as

depth increases (as seen from the size of the circles), indicating that several problems could not be improved beyond a certain point, or that they were not resampled during the pair generation phase. In both these cases, one solution is to allow more LLM calls during phase 1 to allow each problem to be sampled for repair more times. The takeaway here is that more sophisticated fixes for difficult problems can be discovered as we increase the budget of LLM calls during the pair generation phase. The entire evolution of this lineage at different points during pair generation is illustrated in Fig. 12.

## A.2 CODE DIVERSITY

We compute the code diversity score in Fig. 7(b) based on the number of different abstract syntactic sub-trees each code instance produces. First we compute the set of all abstract syntactic sub-trees that the guess code $S_{\text{guess}_j} = \text{AST}(\text{guess}_j)$ and the fix code $S_{\text{fix}_{i,j}} = \text{AST}(\text{fix}_{i,j})$ generates, for every problem/guess $x_j$ in the dataset (in $\mathcal{D}_{\text{test}}$ and every generated fix $i$. For AuPair we use the selected pair $c_i$ in the prompt (a guess/fix pair) to generate a fix $c_i \in \mathcal{A}$ for each AuPair generated in Algorithm 2. Next, we compute the set difference of the generated guess/fix to obtain the unique sub-trees for the code diff $S_{\text{fix}_{i,j}} \setminus S_{\text{guess}_j}$. Then with increase compute $N$, we calculate the unique number of sub-trees generated so far for each problem and compute its average across pairs and problems. The diversity score $\delta$ is for a given compute budget $N$ is written as:

$$\delta = \frac{1}{C\,N\,|\mathcal{D}_{\text{test}}|} \sum_{i=1}^{|\mathcal{D}_{\text{test}}|} \sum_{j=1}^{N} \bigoplus_{i=1} S_{\text{fix}_{i,j}} \setminus S_{\text{guess}_j} \tag{1}$$

this score is normalized by a constant corresponding to the max set size $C =$ $\max_{i,j} \# \left( \bigoplus_{i=1}^{N} S_{\text{fix}_{i,j}} \setminus S_{\text{guess}_j} \right)$. Here we denote $x \in A \setminus B \iff x \in A \setminus B\, x \in A \wedge x \notin B$ as the set difference and use $\bigoplus$ to write the set addition.

Algorithm 4 summarises how we compute the diversity scores:

---

**Algorithm 4** Diversity score computation

---

1: **for** problem $x_j \in \mathcal{D}_{\text{test}}$ **do**
2:     init diversity set of code diffs $\boldsymbol{D}_{0,j} \leftarrow \emptyset$
3:     compute guess AST sub-trees $S_{\text{guess}\,j}$
4:     **for** for every generated fix $i$: **do**
5:         compute fix AST sub-trees $S_{\text{fix}_{i,j}}$
6:         update set of sub-trees $\boldsymbol{D}_{i,j} \leftarrow \boldsymbol{D}_{i-1,j} \bigoplus S_{\text{fix}_{i,j}} \setminus S_{\text{guess}_j}$
7:         count number of unique sub-trees $\delta_{i,j} = \# \boldsymbol{D}_{i,j}$
8:     **end for**
9: **end for**
10: compute its average $\delta = \frac{1}{N|\mathcal{D}_{\text{test}}|} \sum_{i,j} \delta_{i,j}$
11: normalize score $\delta = \delta/C$ with $C = \max_{i,j} \delta_{i,j}$.
    **return $\delta$**

---

## A.3 PROMPTING

There are 2 types of prompts that we use: 1) guess generation prompt, and 2) repair prompt. The guess generation prompt is used during dataset creation, for obtaining the initial guesses for all problems in the dataset. The repair prompt is used throughout the rest of the paper: in the Pair Generation (Phase 1, §2.1 with $k = 32$ random examples) and in the AuPair Extraction (Phase 2, §2.2). The function signature indicates that the function expects a string as an input. The instruction specifies that the final answer is meant to be printed *inside* the function, and that the main function is not meant to be written.

The structure of our repair prompt is as follows: there is a generic instruction at the top, followed by the few-shot examples in the format of question, followed by the guess or bad solution, followed by the fix or good solution. We also add the score achieved by the guess and the fix for the in-context example pairs. Following this, we add the text and initial guess solution for the problem and the

LLM then has to generate a better fix. Note that we do not provide any extra execution feedback in the form of execution traces; this could potentially be explored by future work. Our aim is clear: the pairs indicate a certain type of change and we provide these pairs in context to aid the LLM in generating an improved solution for the given problem. Some different prompting strategies that we tried out were the following:

---

**Guess Generation Prompt**

```
<problem text>
```
Complete the function definition below. Print the final answer in the function. Do not write main. Do not write anything outside the `solve()` function.

```python
def solve(s: str):
    ...
```

---

**Repair Prompt**

You are an experienced software developer.
Look at the question (Q) and solutions below (A).
The main objective is to improve the `solve()` function to answer the question.

Example 1:

(Q): ...
Bad solution code `A(bad)`:

```python
def solve(s: str):
    ...
```

The score of this code is `score(A(bad))` = `<example_guess_score>`.

Good solution code `A(good)`:
The score of this code is `score(A(good))` = `<example_fix_score>`.

```python
def solve(s: str):
    ...
```

:

```
================================================================
```

The main objective is to improve the `solve()` function to answer the question.
(Q): ...
Bad solution code `A(bad)`:

```python
def solve(s: str):
    ...
```

The score of this solution is `score(A(bad))` = `<guess_score>`

Good solution code `A(good)`:
The score of this solution is `score(A(good))` = `100`

---

*Naïve prompting*: only include the problem, guess and fix solutions for the pairs, followed by the problem and guess for the test problem.

*Prompting with instruction only*: include the header instruction followed by the components of the naïve prompting strategy.

*Prompting with instruction and score*: include the elements of 2 above, but in addition, also include the score that each guess/fix received on the corresponding problem's test cases. This is the prompt

that we finally use and the one that gives us better results when compared using the same set of pairs with the previous 2 strategies. An important thing to note here is that we prompt the model with a desired fix score of 100 for the test problem.

We test the three strategies described above on a subset of the CodeForces dataset and report their performance in terms of number of problems solved, in the figure on the right. The results clearly indicate that the final prompting strategy that includes the instruction and score is the best strategy and so we choose it to compose the repair prompt.

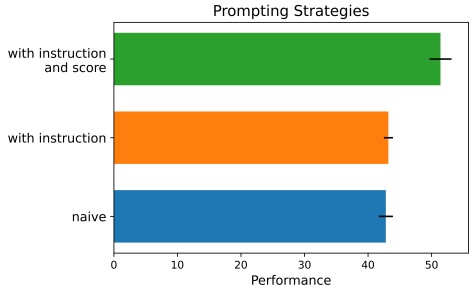

### A.4 CODE EXECUTION

When the LLM generates a fix for any problem, we call the `solve()` function for each test case associated with that problem. We then compare the output with the ground truth and give a partial score corresponding to the proportion of test cases passed by this fix.

An important point to note is that the `solve()` function has to take as input a string, which is then parsed into the correct variables. This formatting requirement is a key reason for the poor initial and best-of-$N$ performance of Gemini-1.5-Pro in Fig. 4. Since the instruction for generating the initial guess is not correctly followed by the model, a lot of guesses end up invariably having incorrect parsing of the input, leading to low scores. A lot of AuPairs extracted using these models, as a result, contain this formatting fix, as we will see in Section A.5.

### A.5 TYPES OF FIXES IN AUPAIRS

We now show some examples of AuPairs and highlight the differences between the guess and fix for each pair. These are a mix of CodeForces pairs collected using the 4 models. The scores achieved by the guess and fix on the corresponding problem's test cases are specified at the top right corner for each example in Fig. 13. We also provide a short description for each type of fix in the caption. The types of pairs discovered using our algorithm cover a large area of potential fixes that can be made to an initial buggy piece of code: from smaller ones like parsing, fixing logical bugs pertaining to indexing errors, variable initialisations, etc., to larger changes like rewriting parts of the code, or even suggesting alternate routes to solve the same problem.

```
score: 0.33
def solve(n: str):
  n = int(n)
  dp = [0] * (n + 1)
  dp[0] = 1
  for i in range(1, n + 1):
    dp[i] = dp[i - 1] * 4
  print(dp[n])
```

```
score: 0.67
def solve(n: str):
  n = int(n)
  dp = [[0 for _ in range(4)] for _ in range(n + 1)]
  dp[0][0] = 1
  for i in range(1, n + 1):
    for j in range(4):
      for k in range(j, 4):
        dp[i][k] = (dp[i][k] + dp[i - 1][j]) %
1000000007
  print(sum(dp[n]))
```

(a) Fix: converts a 1-D dynamic programming solution to 2-D.

```
score: 0.0
def solve(s: str, t: str, k: int):
  n = len(s)
  m = len(t)
  count = 0
  for i in range(n - m + 1):
    match = True
    for j in range(m):
      if abs(i + j - s.find(t[j], i, i + m)) > k:
        match = False
        break
    if match:
      count += 1
  print(count)
```

```
score: 1.0
def solve(s: str):
  n, m, k = map(int, s.split('\n')[0].split())
  S = s.split('\n')[1]
  T = s.split('\n')[2]
  count = 0
  for i in range(n - m + 1):
    match = True
    for j in range(m):
      found = False
      for p in range(max(1, i + j - k), min(n, i + j
+ k) + 1):
        if S[p - 1] == T[j]:
          found = True
          break
      if not found:
        match = False
        break
    if match:
      count += 1
  print(count)
```

(b) Fix: composition of 1) input parsing correction, and 2) logical bug fix.

```
score: 0.5
def solve(s: str):
  n, a, b, c = map(int, s.split())
  if a <= b:
    print(n // a)
  else:
    glass_bottles = n // b
    remaining_money = n - glass_bottles * b
    plastic_bottles = remaining_money // a
    print(glass_bottles + plastic_bottles)
```

```
score: 1.0
def solve(s: str):
  n, a, b, c = map(int, s.split())
  if a <= b:
    print(n // a)
  else:
    glass_bottles = n // b
    remaining_money = n - glass_bottles * b
    plastic_bottles = remaining_money // a
    if remaining_money >= b - c:
      glass_bottles += 1
      remaining_money -= b - c
      plastic_bottles = remaining_money // a
    print(glass_bottles + plastic_bottles)
```

(c) Fix: add an extra condition for edge cases.

```
                                      score: 0.5                                              score: 1.0
def solve(s: str):                                    def solve(s: str):
  n = int(s.split('\n')[0])                             n = int(s.split('\n')[0])
  a = list(map(int, s.split('\n')[1].split()))          a = list(map(int, s.split('\n')[1].split()))
  count = 0                                              count = 0
  for i in range(n):                                    for i in range(n):
    sum = 0                                                sum = 0
    for j in range(i, n):                                 for j in range(i, n):
      sum += a[j]                                           sum += a[j]
      if sum != 0:                                          if sum != 0:
        count += 1                                            count += 1
  print(count)                                             else:
                                                              break
                                                      print(count)
```

(d) Fix: add loop exit condition.

```
                                      score: 0.0                                             score: 0.25
def solve(s: str):                                    def solve(s: str):
  n = int(input())                                      n = int(s.split('\n')[0])
  a = list(map(int, input().split()))                   a = list(map(int, s.split('\n')[1].split()))
  if sum(a) == 0:                                       if sum(a) == 0:
    print("NO")                                           print("NO")
    return                                              else:
  print("YES")                                            print("YES")
  if a[0] != 0:                                            print(n)
    print(1)                                              for i in range(n):
    print(1, n)                                             print(i + 1, i + 1)
  else:
    print(n)
    for i in range(1, n + 1):
      print(i, i)
```

(e) Fix: composition of 1) input parsing correction, and 2) logical bug fix.

```
                                      score: 0.0                                             score: 0.67
def solve(n: int, d: int, a: list[int]):              def solve(s: str):
  balance = 0                                           n, d = map(int, s.split('\n')[0].split())
  bank_visits = 0                                       a = list(map(int, s.split('\n')[1].split()))
  for i in range(n):                                    balance = 0
    balance += a[i]                                     bank_visits = 0
    if balance < 0:                                     for i in range(n):
      if balance + d < 0:                                 balance += a[i]
        print(-1)                                         if balance > d:
        return                                              print(-1)
      bank_visits += 1                                       return
      balance = d + balance                               if a[i] == 0 and balance < 0:
    if balance > d:                                         bank_visits += 1
      print(-1)                                             balance = d
      return                                          print(bank_visits)
  print(bank_visits)
if __name__ == "__main__":
  n, d = map(int, input().split())
  a = list(map(int, input().split()))
  solve(n, d, a)
```

(f) Fix: composition of 1) function signature correction, 2) input parsing correction, and 3) logical bug fix.

```
                                      score: 0.0                                             score: 0.5
def solve(s: str):                                    def solve(s: str):
  n = int(input())                                      n = int(s.split('\n')[0])
  s = input().split()                                   p = list(map(int, s.split('\n')[1].split()))
  p = [int(x) for x in s]                               ans = 1
  ans = 1                                               for i in range(n):
  for i in range(n):                                      ans = ans * p[i] % (10**9 + 7)
    ans = ans * p[i] % (10**9 + 7)                     print(ans)
  print(ans)
```

(g) Fix: input parsing correction.

```
def solve(s: str):                              score: 0.2
  n, p = map(int, s.split())
  dp = [float('inf')] * (n + 1)
  dp[0] = 0
  for i in range(1, n + 1):
    for j in range(32):
      if (1 << j) <= i:
        dp[i] = min(dp[i], dp[i - (1 << j)] + 1)
    if i - p >= 0:
      dp[i] = min(dp[i], dp[i - p] + 1)
  if dp[n] == float('inf'):
    print(-1)
  else:
    print(dp[n])
```

```
def solve(s: str):                              score: 0.6
  n, p = map(int, s.split())
  cnt = 0
  while n > 0:
    if n & 1:
      cnt += 1
    n >>= 1
  if p != 0:
    cnt += (n + p - 1) // p
  print(cnt)
```

(h) Fix: solve problem using bit manipulation instead of dynamic programming.

```
def solve(s: str):                              score: 0.0
  n = int(s.split('\n')[0])
  graph = [[] for _ in range(n + 1)]
  for i in range(1, n):
    u, v = map(int, s.split('\n')[i].split())
    graph[u].append(v)
  visited = [False] * (n + 1)
  count = 0
  for i in range(1, n + 1):
    if not visited[i]:
      count += 1
      dfs(i, graph, visited)
  print(count - 1)
  res = []
  for i in range(1, n + 1):
    if not visited[i]:
      res.append(i)
  print(*res)
def dfs(node, graph, visited):
  visited[node] = True
  for neighbor in graph[node]:
    if not visited[neighbor]:
      dfs(neighbor, graph, visited)
```

```
def solve(s: str):                              score: 0.5
  n = int(s.split('\n')[0])
  graph = [[] for _ in range(n + 1)]
  for i in range(1, n):
    u, v = map(int, s.split('\n')[i].split())
    graph[u].append(v)
  def dfs(node, visited):
    visited[node] = True
    for neighbor in graph[node]:
      if not visited[neighbor]:
        dfs(neighbor, visited)
  count = 0
  visited = [False] * (n + 1)
  for i in range(1, n + 1):
    if not visited[i]:
      count += 1
      dfs(i, visited)
  print(count - 1)
  res = []
  visited = [False] * (n + 1)
  for i in range(1, n + 1):
    if not visited[i]:
      dfs(i, visited)
      res.append(i)
  print(*res)
```

(i) Fix: partial correction to depth-first search graph algorithm.

```
def solve(s: str):                              score: 0.0
  n, p, k = map(int, s.split()[0:3])
  a = list(map(int, s.split()[3:3+n]))
  s = [list(map(int,
s.split()[3+n+i*p:3+n+(i+1)*p])) for i in range(n)]
  people = sorted(enumerate(a), key=lambda x: x[1],
reverse=True)
  max_strength = 0
  for i, (person_index, audience_strength) in
enumerate(people):
    if i == k:
      break
    max_strength += audience_strength
    for j in range(p):
      max_strength_for_position =
max(max_strength_for_position, s[person_index][j])
    max_strength += max_strength_for_position
  print(max_strength)
```

```
def solve(s: str):                              score: 1.0
  n, p, k = map(int, s.split()[0:3])
  a = list(map(int, s.split()[3:3+n]))
  s = [list(map(int,
s.split()[3+n+i*p:3+n+(i+1)*p])) for i in range(n)]
  people = sorted(enumerate(a), key=lambda x: x[1],
reverse=True)
  max_strength = 0
  for i in range(k):
    person_index = people[i][0]
    max_strength += a[person_index]
  for j in range(p):
    best_player_index = -1
    best_player_strength = -1
    for i in range(n):
      if i not in [person[0] for person in
people[:k]]:
        if best_player_strength < s[i][j]:
          best_player_strength = s[i][j]
          best_player_index = i
    max_strength += best_player_strength
  print(max_strength)
```

(j) Fix: rewrite partial solution to pass all test cases.

Figure 13: Examples of AuPairs produced by our algorithm (all 4 models represented above)