# OpenReview forum: "AuPair: Golden Example Pairs for Code Repair"
_ICLR.cc/2025/Conference — Submitted to ICLR 2025_

### Official Review · Reviewer_kjnp · 2024-10-25

**Soundness:** 1
**Presentation:** 2
**Contribution:** 1
**Rating:** 1
**Confidence:** 5

**Summary:**

This paper proposes an approach to self-repair the generation of LLMs for text-to-code tasks. The idea is to pre-compute an example bank of incorrect LLM attempts and its fixes (called AuPairs) from a dataset, then at inference time, select N AuPairs that best match the given utterance to form N 1-shot prompts for the LLMs. The paper shows that its approach out-performs the best-of-N approach on 7 competitive programming datasets using 4 variants of Gemini models. It also demonstrates that the approach generalizes across datasets and variants of Gemini models and scales with inference-time compute budget.

**Strengths:**

+ The paper addresses an important problem

+ The paper is easy to follow

**Weaknesses:**

- The title is confusing. The paper tries to solve text-to-code problems, where the model is given a task description and some test cases. The paper did not solve code repair problems, where the model is typically given a broken program and some test cases to generate the fixed program.

- The evaluation is questionable. The paper did not report results using standard metrics (e.g., pass@k for varies k) and did not provide any comparison with other popular RAG-based approaches. The baseline best-of-N is weak, and its configuration is not clear (zero-shot?).

- The approach may not be practical. To solve each task, it needs to make |C| LLM calls to rank all the candidates in C.

- The paper overclaims that AuPair is a general and domain-agnostic algorithm. It is only evaluated on one domain.

- The paper did not specify several parameters in the algorithms. The claim about composability is not clear.

- The evaluation only shows the results for Gemini variants and missing results for other models.

**Questions:**

1. What is the definition of the score function? Is it the percentage of passing tests?

2. In section 2.1, the discussion is vague. When do you stop generating the candidate pairs (i.e., what do you mean by “large”)? What is the size of C compared to the original training dataset D? What is the value k (the number of randomly sampled examples)? Did you try to control the generation so that each problem in D has a similar representation in C, or did you just do it randomly?

3. In Section 2.2, for each problem in D_{val}, we need to make |C| calls for all the pairs in the candidate list C. This is extremely expensive. Can we use other ways to rank the candidates (e.g., sequence similarity)?

4. In Section 2.2 step 3, why subtracting M by M_k ensures that complementary pairs are generated? Each element in M_k is simply the score of evaluating the pair c_k on D_{val}. The score did not specify which tests passed, so subtracting them did not make sense to me.

5. In Section 2.2, why did you limit to 1-shot in the final prompt? Can’t we use a larger k?

6. I don’t understand the reasons behind picking best-of-N as the baseline. How does your approach perform in standard pass@k metrics? How does your approach compare with other RAG-based techniques?

7. In section 3, can you explain why we may have less than 32 AuPairs given that all datasets are quite large?

8. In Section 3.1, can you provide more details on the best-of-N baseline. Did you use zero-shot?

9. In Section 3.2, I think it is not fair to compare with random 1-shot. Can we compare your approach with other RAG techniques? Also, can you extend your evaluation beyond 1-shot prompts?

10. Can you expand your evaluation to other open source and close source models?

Other minor suggestions: In Figure 1, annotate the diagram with numbers and reference them in the caption. In Figure 2, explain why the score is 0.67 and 1.

---

> ### Author Response · Authors · 2024-11-16
> **Response to reviewer kjnp**
>
> > The title is confusing. The paper tries to solve text-to-code problems, where the model is given a task description and some test cases. The paper did not solve code repair problems.
>
> The paper does attempt to solve the code repair problem (see lines 101-105) as we describe at the beginning of the Approach section: given a problem and its initial guess (i.e. broken program), the task is to generate a fix (i.e. repaired solution), which is scored by the percentage of test cases it passes. The exact repair prompt is shown in the Appendix (section A.2). We will update the text to clarify this point.
>
> > The evaluation is questionable. The paper did not report results using standard metrics (e.g., pass@k for varies k). How does your approach perform in standard pass@k metrics?
>
> We cannot provide fair results for the pass@k metric because the pass@k metric makes an assumption that all $k$ responses from the LLM are i.i.d. generated, whereas our approach produces fixes in a specific order: the first AuPair is more useful than the second, which is more useful than the third, etc; so their success probabilities monotonically decrease as a function of $k$. We will explain this in Section 3.
>
> We report the test case average, or test pass rate, which has also been used in prior work for evaluating LLMs for coding [1, 2, 3].
>
> [1] Measuring Coding Challenge Competence With APPS. Hendrycks et al, 2021.
>
> [2] An Empirical Study of the Non-determinism of ChatGPT in Code Generation. Ouyang et al, 2023.
>
> [3] Benchmarking the Communication Competence of Code Generation for LLMs and LLM Agent. Wu and Fard, 2024.
>
> > How does your approach compare with other RAG-based techniques?
>
> This paper is not related to RAG techniques: we propose an algorithm that devises a fixed set of AuPairs which are used at inference time, as opposed to RAG methods that retrieve relevant documents / knowledge bases at inference time to generate responses. It could be potentially used in conjunction with RAG techniques but then that is orthogonal to the contribution of the paper and is more suitable for future work. In contrast, our set of AuPairs and inference prompts are _fixed_ before any test cases are seen, nothing is retrieved at test time.
>
> > The evaluation only shows the results for Gemini variants and missing results for other models.
>
> We have provided results on 4 models, 2 Gemini models and 2 Gemma models, spanning distinct architectures and meaningfully different levels of capability. But, absolutely, more models are better (as pointed out by Reviewer Yjae as well) so we intend to run more experiments and provide results in a week.
>
> > The approach may not be practical. In Section 2.2, for each problem in D_{val}, we need to make |C| calls for all the pairs in the candidate list C. This is extremely expensive. Can we use other ways to rank the candidates (e.g., sequence similarity)?
>
> Our proposed algorithm discovers AuPairs, which once discovered, can be used to boost performance on several different datasets, as our out-of-distribution generalisation results in Figure 6 indicate. It is important to note that this is a one-time cost that our algorithm has to incur, in return for significant boosts in performance across 6 other datasets using the same fixed set of AuPairs.
>
> Indeed, there could be ways of ranking candidates in less expensive ways such as sequence similarity, distance to closest pair embedding, clustering, etc. In this paper, we chose to rank the candidate pairs by the problems they solved, thereby partitioning the space of pairs in terms of their effect on performance, not the features of the pairs themselves. We did some initial investigations in the direction of clustering pairs by similarity, but the results were not promising enough at the time so we did not pursue it further. This is a good suggestion for future work and we will include it in Section 5.
>
> > I don’t understand the reasons behind picking best-of-N as the baseline.
>
> Our setup involves an inference compute budget of N LLM calls, so we use best-of-N as the baseline since sampling-based methods such as best-of-N and self-consistency have been the most commonly used methods in this setting. As we have also conveyed to reviewer yYRs, we are working on adding an additional self-repair baseline [1] that first gets verbal feedback from the LLM based on the buggy code and then uses that verbal feedback to generate a repaired response.
>
> [1] Is Self-Repair a Silver Bullet for Code Generation? Olausson et al, 2023.

---

> > ### Author Response · Authors · 2024-11-16
> > **Response to reviewer kjnp (continued)**
> >
> > > In Section 3.1, can you provide more details on the best-of-N baseline. Did you use zero-shot?
> >
> > The best-of-N baseline consists of the following formatted according to the repair prompt (see Appendix A.2 for the repair prompt):
> >
> > ```
> > test problem text
> > test problem guess
> > ```
> >
> > The same prompt is then used to generate N responses, out of which the highest-scoring response is selected. Note that here, highest-scoring means the solution that passes the most test cases.
> >
> > On the other hand, the AuPair method consists of the following formatted according to the repair prompt:
> >
> > ```
> > <aupair problem text
> > aupair guess
> > aupair fix>
> >
> > test problem text
> > test problem guess
> > ```
> >
> > This leads to N different prompts for N AuPairs, leading to N responses, out of which, again, the highest-scoring response is selected. Note that due to each AuPair being included as a 1-shot example, we get N different prompts as input to the LLM, compared to the best-of-N setup, where the same prompt is used N times.
> >
> > Finally, the inference compute for best-of-N baseline and AuPair is exactly the same, since both of them make N LLM calls.
> >
> > > The paper did not specify several parameters in the algorithms.
> >
> > We will add a reproducibility section after the main text (according to the ICLR author guidelines) to include all relevant hyperparameters to reproduce our results.
> >
> > > What is the definition of the score function? Is it the percentage of passing tests?
> >
> > Yes, the score function is the average (across problems) of the percentage of passed tests, as we have clarified in our response to an earlier question.
> >
> > > In Section 3.2, I think it is not fair to compare with random 1-shot.
> >
> > We believe random 1-shot is a critical baseline to highlight the importance of AuPairs (which are also evaluated in 1-shot manner) compared to random pairs; the results show that AuPairs obtained by our approach are indeed special (i.e. golden) and even a few AuPairs can recover the performance of a large number of random pairs (see figure 5a).
> >
> > > When do you stop generating the candidate pairs (i.e., what do you mean by “large”)? What is the size of C compared to the original training dataset D?
> >
> > Table 2 in the Appendix shows the exact number of candidate pairs generated for the CodeForces ($|D| = 4000$) and AtCoder ($|D| = 496$) datasets across all 4 models. We stop pair generation when the number of LLM calls crosses a certain threshold (35,000 LLM calls for CodeForces, 10,000 LLM calls for AtCoder), mentioned in Appendix Section A.1.1. Note that this procedure is only used to generate a dataset of (guess, fix) pairs, which could alternatively have been provided beforehand.
> >
> > > Did you try to control the generation so that each problem in D has a similar representation in C, or did you just do it randomly?
> >
> > The generation depends on the ability of the LLM to generate an improved solution for the problem, which may not be the case for difficult problems. As a result, it is not possible to guarantee that each problem in $D$ has a similar representation in $C$. The problems are sampled uniformly from the training dataset for pair generation, and the few-shot examples are sampled randomly from the existing set of candidate pairs.
> >
> > > In Section 2.2 step 3, why subtracting M by M_k ensures that complementary pairs are generated?... subtracting them did not make sense to me.
> >
> > Subtraction of scores ensures that once an AuPair has been selected which solves a set of problems, we select the next AuPair which solves problems from the remaining set (which previous AuPair did not solve), thus ensuring complementarity. This submodular selection algorithm continues till all the AuPairs have been found and gives a set of AuPairs which capture a diverse set of complementary fixes.
> >
> > > In Section 2.2, why did you limit to 1-shot in the final prompt? Can’t we use a larger k?
> >
> > We did not use a larger $k$ because it becomes combinatorially more expensive to figure out the best $k$ shots to put in the prompt. 1-shot keeps our inference algorithm simple and also performs best within the given compute budget.
> >
> > > The paper overclaims that AuPair is a general and domain-agnostic algorithm. It is only evaluated on one domain.
> >
> > Indeed, we got carried away by our enthusiasm here! We’ll tone down all claims to what we have evidence for, and discuss the *potential* for more general applicability in an extended future work section.
> >
> > > In section 3, can you explain why we may have less than 32 AuPairs given that all datasets are quite large?
> >
> > This is due to submodular selection: our submodular algorithm selects AuPairs to solve complementary sets of problems and if the set of candidate pairs (from which AuPairs are selected) does not have enough pairs with complementary properties, there may be fewer than 32 AuPairs produced by the submodular algorithm. Empirically, in our experiments, we have only observed this for the AtCoder dataset with Gemma models (14 for Gemma-9B and 27 for Gemma-27B).

---

> > > ### Author Response · Authors · 2024-11-16
> > > **Response to reviewer kjnp (continued)**
> > >
> > > > Can you expand your evaluation to other open source and close source models?
> > >
> > > We intend to augment our results by evaluating the method on GPT-3.5/4o; would that convince the reviewer that the method is robustly useful and that our claims are justified?
> > >
> > > > What is the value k (the number of randomly sampled examples)?
> > >
> > > We chose $k = 32$ to compose the prompt for pair generation, we will include this detail in the next version.

---

> > > > ### Comment · Reviewer_kjnp · 2024-11-21
> > > >
> > > > Thank you for your response. This does not change my score. I have no more questions.

---

> > > > > ### Author Response · Authors · 2024-11-22
> > > > >
> > > > > Dear reviewer kjnp,
> > > > >
> > > > > In your review, you stated that our paper addresses an important problem, that the method outperforms the baselines, that it generalizes across datasets and models, and that it scales with compute. After such high praise, we were perplexed by the accompanying lowest score at highest confidence -- why would you not rate a paper as clear accept, if it proposes a new, important method that works well across the board?
> > > > >
> > > > > The 16 questions and comments were not making it clear what your primary concern is, but they were mostly clarification questions. We have now addressed all 16 of them in detail, and feel like there should be no remaining misunderstanding. The one open element is the results on an additional model, which we are in the process of running, and will be providing in the next few days (it seems like the new results will confirm the current evidence, just on a broader range of settings). Given that you have no further questions, do you consider all 16 points to be resolved to your satisfaction? Respectfully, what is preventing you from changing your score to an accept?

---

> > > > > > ### Comment · Reviewer_kjnp · 2024-11-26
> > > > > >
> > > > > > Sorry for any misplaced expectations here. The detailed questions that I mentioned were to help you with your editing of any future version (I should have clarified that earlier). The notes in the summary section are not to praise the paper but to rather summarize the paper as per the claims in the paper. Your overall set of responses still fail to address several key concerns I raised in the weakness section including the impracticality of the approach (too many calls for a new query, no clear advantage), inadequate evaluation (weak baselines, did not evaluate on other popular models, datasets---also raised by another reviewer), and unsubstantial claims (domain agnostic, composability claims).

---

> > > > > > > ### Author Response · Authors · 2024-11-29
> > > > > > >
> > > > > > > > The detailed questions that I mentioned were to help you with your editing of any future version (I should have clarified that earlier). The notes in the summary section are not to praise the paper but to rather summarize the paper as per the claims in the paper.
> > > > > > >
> > > > > > > We have revised the draft based on the feedback provided by the reviewers. We provided further information to address the reviewer's concerns, as we believe there may have been some misinterpretations about our work.
> > > > > > >
> > > > > > > > impracticality of the approach (too many calls for a new query, no clear advantage)
> > > > > > >
> > > > > > > This is a misinterpretation; our proposed method at inference time has $N$ LLM calls, which is exactly the same as the baseline ($N = 32$ in our experiments) and is standard in inference-time approaches. Moreover, we show that the cost of finding AuPairs can be amortised effectively since AuPairs generalise well to other datasets (Fig. 6). In fact, it is much more effective than prior work, eg: we show that self-repair with the same number of LLM calls shows inferior performance to AuPair. Our approach shows **clear gains in performance with test-time compute** and it helps with several models, which highlights that it is indeed practical. In fact, we believe that practicality is one major strength of our approach.
> > > > > > >
> > > > > > > > inadequate evaluation (weak baselines, did not evaluate on other popular models, datasets)
> > > > > > >
> > > > > > > We disagree, best-of-$N$ is one of the simplest and strongest baselines in inference-time approaches. We have also added another baseline, self-repair, which has comparable performance to best-of-$N$. It is unclear what baselines the reviewer had in mind.
> > > > > > >
> > > > > > > > unsubstantial claims (domain agnostic, composability claims)
> > > > > > >
> > > > > > > We had a line in the intro and conclusion regarding the domain-agnostic nature of the approach; we have removed it already (please check the updated draft). We stick to the code repair domain throughout the paper and show a large range of results all of which highlight the benefits of using AuPair (sections 3.1-3.8) over other methods, so we are not clear on what the reviewer's concern is.

---

### Official Review · Reviewer_yYRs · 2024-10-26

**Soundness:** 2
**Presentation:** 3
**Contribution:** 2
**Rating:** 5
**Confidence:** 4

**Summary:**

This paper introduces AuPairs, which is designed to enhance the performance of LLMs on auto program repair (APR). The authors utilize each pair of _guess, fix_ in AuPairs as an in-context examplar to improve performance through in-context learning when providing more inference-time compute. The _guess_ is the initial solution generated by the LLM for a coding problem, while the _fix_ is a repaired code for the aforementioned _guess_.

The process of AuPairs is as follows:
First, the initial solution, or _guess_, for all problems in the training set is collected; then, a random sample of $k$ pairs from the candidate pairs (which are initially empty) is used as in-context exemplars to create the _fix_ based on each guess. Subsequently, if the test case pass rate (also defined as score) of the _fix_ exceeds that of the _guess_, the pair _<guess, fix>_ is added to the candidate pairs. Furthermore, if the _fix_ does not pass all cases, it is also converted into a _guess_.

Ultimately, scores for each pair in the candidate pairs are calculated as a one-shot examplar on all guesses in the validation set, and the pairs are selected from high to low based on the average scores. Subsequently, the scores of all remaining pairs' guesses are reduced by the score of the selected pair (with any negative values being set to zero). This method allows for the selection of complementary AuPairs. During testing, each pair in AuPairs is used as a one-shot examplar on the test set.

The contributions of this paper are as follows:
1. AuPairs have outperformed the performance of best of N in multiple scenarios.
2. With an increase in the inference budget, there is an improvement in performance.
3. Good results are achieved across different models and datasets.

**Strengths:**

1. The focal point of AuPairs is well-chosen. By leveraging in-context learning, the method enhances performance without the need for additional training.

2. The approach is ingeniously designed. In the PAIR GENERATION stage, the method creatively reuses imperfect _fix_ by transforming them into _guess_. Additionally, the AuPair EXTRACTION process, which identifies complementary AuPairs by subtracting the scores of selected pairs from those of unselected pairs, is particularly innovative.

3. The method’s effectiveness is validated across multiple models, with comprehensive experimental analysis.

**Weaknesses:**

1. The paper lacks comparisons with alternative baseline methods, as the authors compare only with the simple "best of N" approach. Prior work, such as Self-Repair[1], explores how varying the balance between feedback and repair affects performance under fixed inference attempts. Additionally, it would be valuable to compare AuPairs to methods that provide execution results before resampling fixes. Comparing with other methods that also use a fixed number of inferences would better demonstrate the strengths of the proposed approach.

2. The presentation of the experimental datasets appears overstated. While the authors mention the use of seven datasets, six of these are extracted from CodeContest[2]. While grouping by website categories may be suitable for analysis, I question whether this categorization should be emphasized as a primary contribution. Furthermore, all experiments focus on code generation benchmarks, lacking evaluations on code repair benchmarks like the self-repair section in Livecodebench[3] or the APR section in xCodeEval[4].

3. The conclusions regarding diversity are not sufficiently substantiated. The authors claim that AuPairs provide diversity without a trade-off in performance. However, in Figure 7(b), while diversity scores increase from Gemini-1.5-flash to Gemma-9B to Gemma-27B, performance scores of AuPairs show a decline, suggesting that the improvement in diversity may indeed come at a performance cost too.

4. The figures in the paper could be optimized for information density relative to their visual footprint.

[1] Olausson, Theo X., et al. "Is Self-Repair a Silver Bullet for Code Generation?." ICLR 2023.

[2] Li, Yujia, et al. "Competition-level code generation with alphacode." Science 378.6624 (2022): 1092-1097.

[3] Jain, Naman, et al. "Livecodebench: Holistic and contamination free evaluation of large language models for code." arXiv preprint arXiv:2403.07974 (2024).

[4] Khan, Mohammad Abdullah Matin, et al. "Xcodeeval: An execution-based large scale multilingual multitask benchmark for code understanding, generation, translation and retrieval." ACL 2024.

**Questions:**

1. Why did you choose not to include the execution results of guesses directly within the prompts? Including these results could potentially provide more context and improve the model's performance by refining subsequent guesses.

2. Could you clarify the setup of your baseline method, "best of N"? Specifically, what information is provided during sampling, and what prompt is used? Understanding these details would allow for a clearer comparison with your proposed approach.

---

> ### Author Response · Authors · 2024-11-17
> **Response to reviewer yYRs**
>
> > The paper lacks comparisons with alternative baseline methods…such as Self-Repair[1]
>
> We intend to provide a comparison with self-repair[1]  with the same number of LLM calls at inference in a week.
>
> [1] Is Self-Repair a Silver Bullet for Code Generation? Olausson et al, 2023.
>
> > The presentation of the experimental datasets appears overstated. While the authors mention the use of seven datasets, six of these are extracted from CodeContest[2]... all experiments focus on code generation benchmarks, lacking evaluations on code repair benchmarks like the self-repair section in Livecodebench[3] or the APR section in xCodeEval[4].
>
> Yes, since CodeContests is a collection of several different datasets, 5 of our datasets are borrowed from there (LiveCodeBench and CodeJam used in this paper are not from CodeContests). However, the problems in these individual datasets differ along other axes such as topic coverage, granularity and distribution across difficulty levels, implementation vs mathematically oriented problems, problem description format, to name a few.
>
> Hence, we believe these datasets are still meaningful to assess out-of-distribution generalisation capability of AuPairs since LLMs have been shown to be sensitive to problem formatting [1]. Nonetheless, based on the reviewer's suggestion, we intend to run additional experiments on Livecodebench for repair and report back with results by the end of the discussion period.
>
> [1] Does In-Context Learning Really Learn? Rethinking How Large Language Models Respond and Solve Tasks via In-Context Learning. Long*, Wu* et al, COLM 2024.
>
> > The conclusions regarding diversity are not sufficiently substantiated. The authors claim that AuPairs provide diversity without a trade-off in performance. However, in Figure 7(b), while diversity scores increase from Gemini-1.5-flash to Gemma-9B to Gemma-27B, performance scores of AuPairs show a decline, suggesting that the improvement in diversity may indeed come at a performance cost too.
>
> Figure 7(b) shows the relation between performance and diversity of the generated fixes, with one colour per model. Best-of-N generates diverse but low scoring fixes (except on Flash), while AuPair always generates higher-scoring responses (the star is higher than the square for each model), which can be highly diverse (e.g., when using Gemini-1.5-Pro, top right), or less diverse with Gemma models. In other words, AuPairs never trade off performance, but they also don’t automatically increase diversity.
> We apologise that our caption and original description was not very clear about this, and the plot itself is easy to misread, but the revised version should be much clearer.
>
> > Why did you choose not to include the execution results of guesses directly within the prompts? Including these results could potentially provide more context and improve the model's performance by refining subsequent guesses.
>
> Yes, absolutely, including execution results could potentially improve the model's performance, but as we have described in our setup, we do not expose the model to the actual test cases (we only provide the score), and giving it execution feedback in the context would leak test case information to the model, making this out of scope for this paper. However, this is a great suggestion for future work, so we thank the reviewer for their comment.
>
> > Could you clarify the setup of your baseline method, "best of N"?
>
> The best-of-N baseline consists of the following formatted according to the repair prompt (see Appendix A.2):
>
> ```
> test problem text
> test problem guess
> ```
>
> The same prompt is then used to generate N responses, out of which the highest-scoring response is selected. Note that here, highest-scoring means the solution that passes the most test cases.
>
> On the other hand, the AuPair method consists of the following formatted according to the repair prompt:
>
> ```
> <aupair problem text
> aupair guess
> aupair fix>
>
> test problem text
> test problem guess
> ```
>
> This leads to N different prompts for N AuPairs, leading to N responses, out of which, again, the highest-scoring response is selected. Note that due to each AuPair being included as a 1-shot example, we get N different prompts as input to the LLM, compared to the best-of-N setup, where the same prompt is used N times.
>
> The inference compute for best-of-N baseline and AuPair is exactly the same, since both of them make N LLM calls.

---

> > ### Comment · Reviewer_yYRs · 2024-11-23
> >
> > 1. I look forward to seeing more comparative results on the code repair test sets with other methods! I think the idea of AuPairs is interesting, but the experimental comparisons in the current manuscript are indeed not comprehensive enough.
> >
> > 2. I have participated in numerous algorithm competitions, and in my opinion, the primary difference among platforms (like Coforces/Atcoder/CodeJam) lies in the "format". While some platforms may offer more challenging problems, for easy or medium-level problems, they are generally similar across platforms. Therefore, I do not agree that this constitutes an "out-of-distribution" scenario.
> >
> > 3. Typically, test cases are divided into public test cases and private test cases. It is the responsibility of problem setters to provide public test cases to help participants better understand the problem, especially since language descriptions are sometimes not clear enough, and competition problems are often complex. Private test cases, on the other hand, are used to verify program correctness and cover comprehensive edge cases. Since the CodeContest dataset includes both public and private test cases, I do not think providing public test case results to the model should be considered as "data leakage."

---

> > > ### Author Response · Authors · 2024-11-23
> > >
> > > Thank you for your response! We have updated the draft with additional baseline results.
> > >
> > > > The experimental comparisons in the current manuscript are indeed not comprehensive enough.
> > >
> > > We have now added an extra baseline – self-repair [1]: it is clear from the results, AuPair outperforms self-repair across the board for all 7 datasets and 4 models (has matching performance for only AtCoder Gemma-9B and AtCoder Gemini-1.5-Flash).
> > >
> > > > Since the CodeContest dataset includes both public and private test cases, I do not think providing public test case results to the model should be considered as "data leakage."
> > >
> > > Our setup matches the standard protocol – the test cases that are part of the problem description are the "public" test cases that are available to the model (we do not perform any sort of pruning on the problem description). We do not provide the "private" test cases in the prompt, they are only used for evaluation.
> > >
> > > > The primary difference among platforms (like Coforces/Atcoder/CodeJam) lies in the "format".
> > >
> > > Yes, agreed! However, formatting changes do cause LLMs to fail [2] and hence, form the basis for our OOD evaluation. Despite this fact, our experiments show that AuPairs extracted from one dataset work well across all the other datasets. We believe this result is non-trivial.
> > >
> > > ---
> > >
> > > [1] Is Self-Repair a Silver Bullet for Code Generation? Olausson et al, 2023.
> > >
> > > [2] Does In-Context Learning Really Learn? Rethinking How Large Language Models Respond and Solve Tasks via In-Context Learning. Long*, Wu* et al, COLM 2024.

---

> > > > ### Comment · Reviewer_yYRs · 2024-11-24
> > > >
> > > > Considering the great effort made and the questions solved by the authors during the rebuttal, I'd like to increase my score.

---

> > > > > ### Author Response · Authors · 2024-12-02
> > > > >
> > > > > We thank the reviewer for increasing their score!
> > > > >
> > > > > Just a quick update: as the reviewer suggested, we now have additional results on an additional model, GPT-4o-mini, and an additional dataset, LiveCodeBench (please refer to the [shared update for all reviewers above](https://openreview.net/forum?id=iEdEHPcFeu&noteId=1a4cPnYies). Our results indicate, as before, that **AuPair outperforms all the baselines on the additional model GPT-4o-mini, as well as on the additional dataset, LiveCodeBench**.
> > > > >
> > > > > We would like to thank the reviewer for engaging during the rebuttal process and we hope that this extra set of results addresses the reviewer's concerns.

---

### Official Review · Reviewer_Yjae · 2024-10-27

**Soundness:** 3
**Presentation:** 3
**Contribution:** 2
**Rating:** 5
**Confidence:** 4

**Summary:**

The paper tackles the self-repair task in coding, by synthesizing a set of golden (problem initial guess, fix) pairs, to augment inference-time context when solving test examples. The proposed method shows to be effective across different programming tasks while having the computation overhead controlled.

**Strengths:**

Strengths
1. The paper tackles an interesting problem – coding repairs – and experiment on a wide range of datasets to provide its comphrensiveness and effectiveness.

2. The proposed method shows to be effective on a series of in-domain and out-of-domain, on weaker Gemma and stronger Gemini models, partially demonstrating its effectiveness.

3. The controlled compute budget (N LLM calls) alleviate worries in gaining increase with more compute.

**Weaknesses:**

1. The experimented setting somewhat already “reveals the answer”: as described in the paper “given a coding problem, its test cases, and …”, the test cases serve as a canonical correctness checker for model generations – one only need to execute model-generated code over the test cases to know if it’s correct or not. AFAIK the test cases should be assumed unknown at any point during the generation.

2. Incomprehensive model selection: this paper experiments with Gemma and Gemini models in various sizes. However, neither Gemma nor Gemini are known to perform the best on coding tasks, their coding alternatives (e.g., CodeGemma) and better alternatives (CodeLlama, DeepseekCoder) should be considered to prove the effectiveness of the method. Otherwise, the paper should clearly state that the proposed method only applies to text-oriented LMs or weak coding LMs.

3. It is unclear (from section 3) what evaluation metric is used in this paper, especially given that rigorous evaluation is important for coding tasks. From the description, it seems that “the number of unit tests passed” is the main metric. First, writing-wise, it’d be good to highlight (e.g., bold) the metric name to clearly inform the readers. Second, design choice wise, the “number of tests passed” do not correlate with the correctness extent of a program. A program should pass all test cases to prove its correctness; and “number of tests passed” will be more affect by the distribution/sufficiency of test cases (which analysis is not available in the paper) than the actual correctness of the model-generated program.

**Questions:**

1. Could you explain how the method control the N LLM call compute budget? Because the experiments seem to control the sampling baseline to sample only N responses; but for the proposed method, it involves N AuPairs and then N fixes, so that would cause the compute to be 2N?

2. It is unclear what the green and blue blocks mean in Figure 1; similarly, what the purple block mean in Figure 3. Thus I’m not sure if I fully understand the figures.

2. I am curious on the types of fixes model learn to perform, have you observed any major catefogories, e.g., “fixing variable name copy errors”, “fixing logical computing errors”, etc.

4. As mentioned in the paper, AuPair is created on a held-out validation set. How would the proposed method apply when no held-out validations are available?

---

> ### Author Response · Authors · 2024-11-16
> **Response to reviewer Yjae**
>
> > The experimented setting somewhat already “reveals the answer”: as described in the paper “given a coding problem, its test cases, and …” …  test cases should be assumed unknown at any point during the generation.
>
> The test cases are not provided to the model as part of the prompt. The part "given a coding problem, its test cases, and …" is used to refer to the full setup where the test cases are  _not_ provided during response generation, they are only used for evaluation. We will update the text to clarify this.
>
> > this paper experiments with Gemma and Gemini models in various sizes…better alternatives should be considered
>
> We intend to provide additional results for GPT-3.5/4o next week (and will follow up on this as we get results).
>
> > It is unclear (from section 3) what evaluation metric is used in this paper… A program should pass all test cases to prove its correctness…
>
> We report the test case average, or test pass rate, which has also been used in prior work for evaluating LLMs for coding [1, 2, 3]. We will highlight the metric for easier understanding.
>
> We will also report the number of problems with all test cases passed: this is less granular but addresses the potential risk of bias pointed out.
>
> [1] Measuring Coding Challenge Competence With APPS. Hendrycks et al, 2021. \
> [2] An Empirical Study of the Non-determinism of ChatGPT in Code Generation. Ouyang et al, 2023. \
> [3] Benchmarking the Communication Competence of Code Generation for LLMs and LLM Agent. Wu and Fard, 2024.
>
> > Sampling baseline sample only N responses; but for the proposed method, it involves N AuPairs and then N fixes, so that would cause the compute to be 2N?
>
> Both baseline and our proposed approach use N LLM calls. The only difference between baseline and our approach is in the prompt that is passed to the LLM. In baseline, all the N LLM calls use the same prompt containing the problem text and guess code:
>
> ```
> test problem text
> test problem guess
> ```
>
> In AuPair, each of the N LLM calls contains an AuPair as the 1-shot example, formatted according to the repair prompt (Appendix A.2):
>
> ```
> <aupair problem text
> aupair guess
> aupair fix>
>
> test problem text
> test problem guess
> ```
>
> To be extra clear: AuPair uses N calls, not 2N, just like the baselines we compare to.
>
> > It is unclear what the green and blue blocks mean in Figure 1; similarly, what the purple block mean in Figure 3.
>
> The blue blocks in Figures 1 and 3 refer to "guesses" from the train dataset and the green blocks are "fixes". Note that the train dataset contains blue guesses, and the LLM after repair generates green fixes. Pairs consist of guesses and fixes both, so they are represented as the combination of the blue and green.
>
> The purple blocks in Figure 3 refer to guesses from the validation dataset. In both Figures 1 and 3, the pairs have blue and green blocks, indicating that pairs are collected and AuPairs are extracted only on the training dataset.
>
> > I am curious on the types of fixes model learn to perform, have you observed any major categories, e.g., “fixing variable name copy errors”, “fixing logical computing errors”, etc.
>
> Section A.5 in the Appendix contains examples of 10 types of fixes, along with a short description of each type of fix. We have observed categories such as adding extra if conditions for edge cases, fixing input parsing, adding loop exit condition, fixing logical bugs, to name a few.
>
> > As mentioned in the paper, AuPair is created on a held-out validation set. How would the proposed method apply when no held-out validations are available?
>
> In that case, we split any given dataset for which we want to generate AuPairs, into training, validation, and test datasets. So if the validation dataset is not provided separately, we construct this split from our side to get a validation dataset.

---

> > ### Author Response · Authors · 2024-12-02
> >
> > We thank the reviewer for their feedback and would like to highlight the key results and takeaways from the discussion period:
> >
> > - **Additional baseline**: We have now added an extra baseline – self-repair [1]. It is clear from the results, *AuPair outperforms self-repair across all datasets and models* (has matching performance for only AtCoder Gemma-9B and AtCoder Gemini-1.5-Flash).
> >
> > - **Additional model**: We now have results using GPT-4o-mini on AtCoder:
> >
> > | Baseline | Test pass rate |
> > | --- | --- |
> > | Initial | 27.83% |
> > | Best-of-$N$ | 18.49% |
> > | Self-repair | 31.29% |
> > | **AuPair** | **41.4%** |
> >
> > As is evident from these results, all our trends from the other models hold: *AuPair outperforms all baselines even on GPT-4o-mini*.
> >
> > - **Additional dataset**: We have added results for repair on LiveCodeBench in the appendix of the updated draft (Section A.1.4); the results clearly indicate that all the trends we have observed in the original set of experiments hold: *AuPair exhibits superior performance to the baselines across all models.*
> >
> > - **Additional metric**: We have also reported the percentage of fully solved problems as an additional metric in Fig. 11 in the appendix. All the trends that we observed for the earlier metric, test pass rate, hold for this stricter metric as well.
> >
> > - **Text revisions**: We have updated the figure captions to clarify the approach and added an extra pair generation algorithm) in the appendix; we hope this gives a clear idea of the steps involved in the AuPair algorithm.
> >
> > Given that the extended discussion period is drawing to an end, we would like to request the reviewer to reconsider their score, especially in light of all the additional results and clarifications we have provided that clearly show that our proposed approach outperforms all baselines across a large number of models and datasets.
> >
> > ---
> >
> > [1] Is Self-Repair a Silver Bullet for Code Generation? Olausson et al, 2023.

---

### Official Review · Reviewer_V5Ww · 2024-11-01

**Soundness:** 2
**Presentation:** 2
**Contribution:** 3
**Rating:** 6
**Confidence:** 2

**Summary:**

The paper deals with the problem of self-repairing the answers to an LLM’s responses. The authors claim the proposal of a general-purpose, domain-independent algorithm (according to lines 079--081) for improving an LLM’s responses.

**Strengths:**

The paper conducts an interesting empirical comparison of the proposed technique for self-repair in code repairing.

**Weaknesses:**

- In line 079, the authors claim that they propose a general-purpose, domain-independent algorithm for self-repairing LLMs. However, later, in line 082, the authors state that they focus on code repairing. I find this very confusing. The authors must either give a second use case to support the claim about domain-independence, or remove the claim made in lines 079—081 and exclusively present code repairing as their aim.
- Continuing with my previous questions, the unit tests are an essential part of the pipeline – according to line 072, the fixes are evaluated on the unit tests. However, unit tests or other kinds of assertion statements are, in general, missing in the setting of detecting hallucinations (e.g., factual mistakes) in the responses of an LLM (e.g., as it is the case when asking a VLM questions on a specific image). Hence, I do not see how the proposed framework can be applied to other domains. Can the authors please clarify?
- In line 143, the authors state that a set \mathcal{C} of candidate guess-fix pairs is extracted using the methodology described in Figure 1. However, Figure 1 is high-level, giving no concrete details. Can the authors please be more specific of how the extraction algorithm works?
- In line 072, it is unclear how the (optional) examples are used during the sampling process and in general, what is the purpose of these optional examples in the overall pipeline. Can the authors please provide further details?
- In line 160, the authors state that a validation set is needed during the AuPair extraction process. However, in lines 082—083, this part is missing. Can the authors be very specific on what is exactly is needed as an input to their algorithm?

The above issues make it difficult to the reader understand the paper and its contributions. However, I am happy to revise my score if the authors address/answer my concerns/questions.

**Questions:**

Please see my comments/questions in the above field. In addition:
- Please provide a concrete running example, potentially extending/giving more information on the example in Figure 2. Please clarify how the unit tests look like, how the examples look like, how the candidate guess-fix pairs look like, how the validation set looks like. Even a toy example that demonstrates these notions would enhance readability a lot.
- Please discuss how the method might be adapted or generalized to work in domains without clear unit tests or ground truth evaluations.
- Please provide a more detailed algorithmic description or pseudocode for the extraction process, in addition to the high-level overview in Figure 1.
- Please provide a clear list of all inputs required by the algorithm, including the validation set, and explain how each input is used in the process.
- Please position your work against SOTA on detecting/mitigating hallucinations in LLMs and VLMs. One indicative reference is "HALC: Object Hallucination Reduction via Adaptive Focal-Contrast Decoding", ICML 2024, which studies the problem of detecting hallucinations in VLMs.

---

> ### Author Response · Authors · 2024-11-16
> **Response to reviewer V5Ww**
>
> > In line 079, the authors claim that they propose a general-purpose, domain-independent algorithm for self-repairing LLMs. However, later, in line 082, the authors state that they focus on code repairing.
>
> We wanted to emphasise that our _algorithm_ is domain-independent but in this paper, we focus on the code-repair task as an evaluation for our proposed algorithm. We got carried away by our enthusiasm while writing but we'll tone down this sentence, thanks for pointing this out!
>
> > The unit tests are an essential part of the pipeline…I do not see how the proposed framework can be applied to other domains. Can the authors please clarify?
>
> Intuitively, the unit tests act as grounded feedback for our algorithm which is used to drive the selection of AuPair in the right direction. However, it is not the only way of feedback. As mentioned in the Conclusion section (line 538), we have talked about using feedback from a reward model or a critic [1, 2]. In this case, the scores that we currently obtain from evaluation on unit tests can be replaced by the scores produced by the critic without having to change the AuPair algorithm.
>
> [1] CRITIC: Large language models can self-correct with tool-interactive critiquing. Gou et al, 2024 \
> [2] CodeRL: Mastering Code Generation through Pretrained Models and Deep Reinforcement Learning. Le*, Wang* et al, 2022.
>
> > Please provide a clear list of all inputs required by the algorithm, including the validation set, and explain how each input is used in the process.
>
> In this work, we have two phases: (1) Candidate pair dataset generation consisting of guess-fix pairs (2) AuPair selection algorithm. AuPair selection algorithm is concretely expressed in Algorithms 1 and 2 with exact inputs and outputs, which is our core contribution.
>
> Candidate pair dataset generation is explained in detail below and we'll include it in the Appendix. However, note that if a pair dataset is given with guess-fix samples, this stage is not required.
>
> Full pipeline with input-output listed in brackets: \
> Given: coding dataset consisting of problem description and unit tests
>
> Phase 1:
> - Construct guess dataset: generate initial guess (using guess generation prompt in Appendix A.2) for each problem in the input coding dataset, and the score for this initial guess as the percentage of unit tests passed by the initial guess (input: coding dataset containing problem descriptions and unit tests, output: guess dataset containing problem descriptions, unit tests, initial LLM-generated guesses, scores for guesses)
>
> - Train / val / test split from guess dataset (input: guess dataset, output: train dataset, val dataset, test dataset)
>
> - Generate large set of candidate pairs using the train dataset (input: train dataset, output: candidate pairs, see Figure 1)
>
> Phase 2:
>
> - Extract AuPairs from the candidate pairs by evaluating on the validation dataset (input: candidate pairs, val dataset, output: AuPairs, see Figure 3, Algorithms 1 and 2)
>
> Inference:
>
> - Evaluate the impact of using AuPairs on the test dataset (input: AuPairs, test dataset, output: final score)
>
> > Figure 1 is high-level, giving no concrete details. Can the authors please be more specific of how the extraction algorithm works?
>
> We give the details of how to generate pairs in the caption for Figure 1, but we can flesh this out in the main text to make it easier to understand. The pair generation phase proceeds as follows:
>
> 1. We begin with the curated dataset that consists of the problem, its initial guess generated by the LLM, and the score of this initial guess. This score is computed as the percentage of passed unit tests for that problem by the guess.
>
> 2. Next, a problem and its guess are sampled from this dataset and the repair prompt is composed (the exact prompt is given in the Appendix (Section A.2).
>
> 3. This repair prompt is then passed as input to the LLM which generates a repaired solution, or a fix.
>
> 4. Next, the score of that fix is calculated in the same way as the score of the guess. If the fix score is higher than a guess score, this pair of (guess, fix) along with their scores, and the problem and unit tests is stored in the set of candidate pairs.
>
> 5. Furthermore, if this fix does not solve all unit tests, it is added back into the dataset as a new guess, since there is further scope for improvement.

---

> > ### Author Response · Authors · 2024-11-16
> > **Response to reviewer V5Ww (continued)**
> >
> > > In line 072, it is unclear how the (optional) examples are used during the sampling process and in general, what is the purpose of these optional examples in the overall pipeline. Can the authors please provide further details?
> >
> > The examples we are referring to in line 072 are the pairs of guesses and fixes that can be provided as in-context along with the [problem, guess] for pair generation. The purpose of these examples in this pair generation phase is to leverage the in-context learning capability of LLMs so that potentially better fixes, and consequently better pairs, could be generated.
> >
> > We included the word "optional" (line 102) to encapsulate different baselines and approaches used in this work which include additional in-context examples along with [problem, guess] to solve the code-repair task. We'll remove it to avoid any further confusion.
> >
> > > In line 160, the authors state that a validation set is needed during the AuPair extraction process. However, in lines 082—083, this part is missing.
> >
> > In line 82-83, we just define the code-repair task (and what its inputs and outputs are). AuPair extraction algorithm requiring a held-out validation dataset is not connected to the definition of the code-repair task, but it is part of the solution to this task. As the reviewer correctly pointed out, the solution algorithm requires a validation set to discover AuPairs. If the validation dataset is not provided separately, we construct this split from our side to get a validation dataset (also mentioned in response to above question and in line ). This step is important to establish the quality of each candidate pair on unseen validation examples.
> >
> > > Please provide a more detailed algorithmic description or pseudocode for the extraction process, in addition to the high-level overview in Figure 1.
> >
> > Figure 1 refers to the pair generation process, we will add an extra algorithm that describes pair generation in the upcoming revision for better understanding. The algorithm for the AuPair extraction process is already provided (refer to Algorithm 1 and 2).
> >
> > > Please position your work against SOTA on detecting/mitigating hallucinations in LLMs and VLMs. One indicative reference is "HALC: Object Hallucination Reduction via Adaptive Focal-Contrast Decoding", ICML 2024, which studies the problem of detecting hallucinations in VLMs.
> >
> > Thank you for pointing us to this reference! Indeed, it turns out that this setting has potential overlap with our method: the visual grounding and FOV-based detectors that HALC uses during inference could be seen as analogues the unit tests in our setting, and hence an AuPair-like method could potentially be transferred to the visual hallucination domain. We will cite and discuss this in the future work section.

---

> ### Author Response · Authors · 2024-12-02
>
> We thank the reviewer for their feedback and would like to highlight the key results and takeaways from the discussion period:
>
> - **Additional baseline**: We have now added an extra baseline – self-repair [1]. It is clear from the results, *AuPair outperforms self-repair across all datasets and models* (has matching performance for only AtCoder Gemma-9B and AtCoder Gemini-1.5-Flash).
>
> - **Additional model**: We now have results using GPT-4o-mini on AtCoder:
>
> | Baseline | Test pass rate |
> | --- | --- |
> | Initial | 27.83% |
> | Best-of-$N$ | 18.49% |
> | Self-repair | 31.29% |
> | **AuPair** | **41.4%** |
>
> As is evident from these results, all our trends from the other models hold: *AuPair outperforms all baselines even on GPT-4o-mini*.
>
> - **Additional dataset**: We have added results for repair on LiveCodeBench in the appendix of the updated draft (Section A.1.4); the results clearly indicate that all the trends we have observed in the original set of experiments hold: *AuPair exhibits superior performance to the baselines across all models.*
>
> - **Additional metric**: We have also reported the percentage of fully solved problems as an additional metric in Fig. 11 in the appendix. All the trends that we observed for the earlier metric, test pass rate, hold for this stricter metric as well.
>
> - **Text revisions**: We have updated the figure captions to clarify the approach and added an extra pair generation algorithm) in the appendix; we hope this gives a clear idea of the steps involved in the AuPair algorithm.
>
> Given that the extended discussion period is drawing to an end, we would like to request the reviewer to reconsider their score, especially in light of all the additional results and clarifications we have provided that clearly show that our proposed approach outperforms all baselines across a large number of models and datasets.
>
> ---
>
> [1] Is Self-Repair a Silver Bullet for Code Generation? Olausson et al, 2023.

---

### Author Response · Authors · 2024-11-23
**Update for all reviewers**

We thank the reviewers for their helpful feedback on our paper! We have uploaded a new draft (updates below). Since the rebuttal deadline is approaching soon (Nov 26, 2024), we would like to request the reviewers to reconsider their scores given the additional evidence and clarifications that we have provided.

- **Additional baseline**: We have now added an extra baseline (as requested by reviewers yYRs and kjnp) – self-repair [1]. *It is clear from the results, AuPair outperforms self-repair across all datasets and models* (has matching performance for only AtCoder Gemma-9B and AtCoder Gemini-1.5-Flash).

- **Additional metric**: We have also reported the percentage of fully solved problems as an additional metric (as requested by reviewer Yjae) in Fig. 11 in the appendix. *All the trends that we observed for the earlier metric, test pass rate, hold for this stricter metric as well*.

- **Text revisions**: We have updated the figure captions to clarify the approach and added an extra pair generation algorithm (as requested by reviewer V5Ww) in the appendix; *we hope this gives a clear idea of the steps involved in the AuPair algorithm*.

---

[1] Is Self-Repair a Silver Bullet for Code Generation? Olausson et al, 2023.

---

### Author Response · Authors · 2024-12-02

We would like to give an additional update to all the reviewers, we have the following new results:

- **Additional model**: We now have results using GPT-4o-mini on AtCoder:

| Baseline | Test pass rate |
| ----- | ----- |
| Initial | 27.83% |
| Best-of-$N$ | 18.49% |
| Self-repair | 31.29% |
| **AuPair** | **41.4%** |

As is evident from these results, our trends from the other models hold: **AuPair outperforms all baselines even on GPT-4o-mini**.


- **Additional dataset**: As suggested by reviewer yYRs, we have added results for repair on LiveCodeBench in the appendix of the updated draft (Section A.1.4); the results clearly indicate that all the trends we have observed in the original set of experiments hold: **AuPair exhibits superior performance to the baselines across all models.**

Given that the extended discussion period is drawing to an end, we would like to request the reviewers to reconsider their scores, especially in light of all the additional results and clarifications we have provided that clearly show that our proposed approach outperforms all baselines across a large number of models and datasets.

---

### Meta-Review · Area_Chair_pX4r · 2024-12-22

**Metareview:**

This paper considers the problem of code generation using large language models (LLMs) and takes a self-repair approach. The key idea is to create a set of incorrect LLM attempts and fixes to augment them as context at inference time while solving test problems. Experiments on multiple programming datasets using variants of Gemini models shows good improvement over best-of-N approach.

The paper looks at an important problem and the proposed approach is promising. The reviewers' raised a number of questions including some key claims and the rebuttal tried to address most of them. One reviewer who was negative seems to be harsh in my opinion. After reading all the reviews, discussion, and taking a look at the paper myself, I felt the paper is borderline and could be improved in two main directions.
1. The paper could consider more code LLMs of varying quality to demonstrate the improvement as a function of the quality of model. The efficacy of the approach is not clear from this perspective.
2. There is a good amount of work on test-driven code generation/repair using LLMs (there is work in both ML and software engineering venues). Since the current approach requires additional LLM calls at test-time, it will be good to compare with these approaches to evaluate the performance difference and compute cost overhead to achieve improvement. The paper currently focuses only on best-of-N and self-repair as baselines. Another useful baseline is passing one or more unsolved test cases as part of the prompt.

Therefore, I'm recommending to reject this paper and strongly encourage the authors' to improve the paper based on the feedback from reviewers' for resubmission.

**Additional Comments On Reviewer Discussion:**

The reviewers' raised a number of questions including some key claims and the rebuttal tried to address most of them. Only two reviewers' engaged in the discussion. One reviewer who was negative seems to be harsh in my opinion.

After reading all the reviews, discussion, and taking a look at the paper myself, I felt the paper is borderline and there is room for improvement.

---

### Decision · Program_Chairs · 2025-01-22

Reject